# Induced surface fluxes: A new framework for attributing Arctic sea ice volume balance biases to specific model errors

Alex West[1], Mat Collins[2], Ed Blockley[1], Jeff Ridley[1], Alejandro Bodas-Salcedo[1]

[1]Met Office Hadley Centre, FitzRoy Road, Exeter, EX1 3PB
[2]Centre for Engineering, Mathematics and Physical Sciences, University of Exeter, Exeter, EX4 4SB, UK

*Correspondence to*: Alex E. West (alex.west@metoffice.gov.uk)

**Abstract.** A new framework is presented for analysing the proximate causes of model Arctic sea ice biases, demonstrated with the CMIP5 model HadGEM2-ES. In this framework the Arctic sea ice volume is treated as a consequence of the integrated surface energy balance, via the volume balance. A simple model allows the local dependence of the surface flux on specific model variables to be described, as a function of time and space. When these are combined with reference datasets, it is possible to estimate the surface flux bias induced by the model bias in each variable. The method allows the role of the surface albedo and ice thickness-growth feedbacks in sea ice volume balance biases to be quantified, along with the roles of model bias in variables not directly related to the sea ice volume. It shows biases in the HadGEM2-ES sea ice volume simulation to be due to a bias in spring surface melt onset date, partly countered by a bias in winter downwelling longwave radiation. The framework is applicable in principle to any model and has the potential to greatly improve understanding of the reasons for ensemble spread in modelled sea ice state. A secondary finding is that observational uncertainty is the largest cause of uncertainty in the induced surface flux bias calculation.

## 1.   Introduction

The Arctic sea ice cover has witnessed rapid change during the past 30 years, with a decline in September extent of $1.05 \times 10^6$ km$^2$ / decade from 1986 to 2015 (HadISST1.2, Rayner et al 2003). In association with the changes in extent, Arctic sea ice thinning has been observed from submarine and satellite data (Rothrock et al, 2008, Lindsay and Schweiger, 2015). Arctic sea ice has also become younger on average as older ice has been lost (Maslanik et al, 2011), the onset of summer melt has become earlier in the year (Markus et al, 2009) and the onset of winter freezing has become later (Stammerjohn et al, 2012).

The changes have focussed interest on model projections of Arctic sea ice, the loss of which influences the climate directly through increased absorption of shortwave (SW) radiation during summer and through greater release of heat from the ocean to the atmosphere during winter (Stroeve et al, 2012b). However, substantial spread remains in model simulations of present-day Arctic sea ice and of the long-term rate of decline under climate change (Stroeve et al, 2012a). The causes of this spread are poorly understood, resulting in large uncertainty in future projections of Arctic sea ice.

Evaluating sea ice extent or volume with respect to reference datasets shows that some models reproduce present-day sea ice state more accurately than others (e.g. Wang and Overland, 2012; Massonnet et al, 2012; Shu et al, 2015). However, an accurate simulation of sea ice extent and volume under the present-day climate does not necessarily imply an accurate future projection of sea ice change, as a correct simulation can be

obtained by accident due to cancelling model errors. Sea ice extent in particular is known to be a very unsuitable
metric for diagnosing model performance due to its high internal variability (Notz, 2015; Swart et al, 2015).
Hence there is a need to better understand the drivers which lead a model to simulate a given Arctic sea ice
state.
This study presents a new framework (the induced surface flux, or ISF, framework) to improve understanding of
sea ice model bias, by identifying proximate drivers of model bias in sea ice volume balance. The framework is
motivated in the following way. Changes in sea ice volume are driven by the sea ice volume balance. Sea ice
volume balance in turn arises from the surface and basal energy balance, as given a roughly constant ice density,
the ice volume is to first order proportional to the energy required to melt the ice out. Basal melting in the
interior ice pack has been shown to result, in the main, from direct solar heating of the ocean (e.g Maykut and
McPhee, 1995), while basal freezing derives principally from conduction of energy upward through the ice
(Perovich and Elder, 2002). This implies that the total downwards surface energy flux (surface flux) contains the
principal sources and sinks of energy for the sea ice volume balance on an Arctic-wide scale. However, a
complex two-way relationship exists between sea ice thickness and surface energy balance, via the surface
temperature and surface albedo, giving rise to the thickness-growth feedback (Bitz and Roe, 2004) and the
surface albedo feedback (Bitz, 2008), both of which exert first-order control on the sea ice state. Hence many
components of the SEB cannot be viewed as independent of the sea ice state, but are directly affected by it.
Therefore, there is a need to separate surface flux biases that are caused by the sea ice state (representing
feedbacks of the sea ice volume balance) from those that are not (representing forcings on the sea ice volume
balance).
In the ISF framework, the total downwards surface energy flux is expressed as an explicit function of key Arctic
climate variables, allowing the dependence of modelled surface flux on each variable to be described. Hence,
using reference datasets (observational estimates or reanalyses) for the climate variables, the model bias in
surface flux induced by each climate variable can be estimated. In this way, biases in ice growth and melt over
the course of the year are attributed, via the surface flux, to biases in specific model quantities. The method
allows the contributions to model biases in ice growth and melt caused by the sea ice albedo feedback, the ice
thickness-growth feedback, and various external factors, or 'forcings', to be separately quantified. In this way it
can be seen how model biases in the external forcings drive model bias in the sea ice volume balance, offering a
valuable tool for setting sea ice state biases in context, and for understanding spread in sea ice simulation within
multi-model ensembles.
In more detail, the ISF framework works by expressing the total downward surface energy flux $F_{sfc}$, at each
point in time $t$ and space $x$, as an explicit function $g_{x,t}$ of quasi-independent climate variables $v_i$. The variables
are quasi-independent in the sense that while they affect each other on timescales varying from days to months,
they affect the surface flux instantaneously. Hence by taking partial derivatives, the dependence of the surface
flux on each variable can be separately expressed at each point in time and space ($\left[\partial g_{x,t}/\partial v_i\right]^{MODEL}$). Given
an estimate of the model bias in any variable via a reference dataset $\left(v_{i,x,t}^{MODEL} - v_{i,x,t}^{REFERENCE}\right)$, the field of

surface flux dependence can be multiplied through to produce an estimate of the surface flux bias induced (instantaneously) by the model bias in that variable, as a function of space and time.

This method has two key strengths; firstly, that the fields of induced surface flux bias (ISF bias) can be averaged in time or space to determine the large-scale effects of particular model biases, bypassing nonlinearities in surface flux dependence. Secondly, due to the quasi-independence of the variables, the effects of each on the surface flux are separated, such that the sum of the ISF biases theoretically approaches the total, true, model surface flux bias. In this way, the instantaneous, or proximate, causes of the model surface flux bias, and hence sea ice volume balance bias, can be separated and quantified.

The analysis is applied to the four members of the historical ensemble of the coupled CMIP5 model HadGEM2-ES. This model simulates anomalously low annual minimum ice extent, as well as ice volume that is both too low in the annual mean and too amplified in the seasonal cycle, a similar behaviour to that identified by Shu et al (2015) in the CMIP5 ensemble mean. A variety of reference datasets are used to assess model biases, to demonstrate the large observational uncertainties present in the Arctic, and their effect on our ability to attribute sea ice volume balance bias with the ISF framework.

The paper is structured as follows. In Section 2, the HadGEM2-ES model and the reference datasets used are described in turn. In Section 3, the sea ice and surface radiation simulations of HadGEM2-ES are evaluated; in demonstrating the sea ice volume balance biases of HadGEM2-ES, as well as possible drivers, this motivates the following analysis. In Section 4, the ISF framework is described in more detail, and examples shown. In section 5 the ISF analysis is applied to HadGEM2-ES, allowing the role of each model bias identified in Section 3 in causing sea ice volume balance bias to be quantified. In Section 6 the implications of the results are discussed, in particular the mechanisms by which the identified external drivers determine the modelled sea ice state, and the likely drivers behind the corresponding model biases. Conclusions are presented in section 7.

## 2. Model and reference datasets

### 2.1 The HadGEM2-ES model

HadGEM2-ES is a coupled climate model employing additional components to simulate terrestrial and oceanic ecosystems, and tropospheric chemistry (Collins et al, 2011). It is part of the HadGEM2 'family', a collection of models that all use the HadGEM2-AO coupled atmosphere-ocean system. HadGEM2-AO is developed from HadGEM1 (Johns et al, 2006), a coupled atmosphere-ocean model whose sea ice extent simulation was recognised as being among the closest to observations out of the CMIP3 ensemble models (Wang and Overland, 2009). While the atmospheric and ocean components of HadGEM2-ES contain a large number of improvements relative to HadGEM1, many of these targeted at improving simulations of tropical weather, the sea ice component is very similar to that of HadGEM1 (except for three minor differences, summarised in Martin et al, 2011, table A4).

A fundamental feature of the sea ice component of HadGEM2-ES is the sub-gridscale sea ice thickness distribution (Thorndike et al, 1975). In this formulation, ice in each grid cell is separated into five thickness

categories with boundaries at 0, 0.6m, 1.4m, 2.4m, 3.6m and 20m, each with its own area, thermodynamics and
surface exchange calculations. Another key aspect of the sea ice model is the zero-layer thermodynamics
scheme (appendix to Semtner, 1976), in which the sea ice surface temperature responds instantaneously to
changes in forcing, conduction is uniform within the ice and snow column, and neither sea ice nor overlying
snow have heat capacity (sensible heat storage is parameterised in the top 10cm of the snow-ice column during
surface exchange calculations, to aid stability).
The HadGEM2-ES sea ice model also includes elastic-viscous-plastic sea ice dynamics (Hunke and Dukowicz,
1997) and incremental remapping (Lipscomb and Hunke, 2004). Most sea ice processes are calculated in the
ocean model, but the surface energy balance (SEB) calculations are carried out in the atmosphere model, which
passes top melting flux and conductive heat flux to the ocean model as forcing for the remaining components. A
more complete description of the sea ice component can be found in McLaren et al (2006).
This study uses the four ensemble members of the CMIP5 historical experiment of HadGEM2-ES, forced with
observed solar, volcanic and anthropogenic forcing from 1860 to 2005. The period 1980-1999 is used for the
model evaluation, chosen to be close to the end of the period of the historical experiments so as to be. recent
enough to allow the use of a reasonable range of observational data. All analysis is carried out with data
restricted to the Arctic Ocean region, defined as the area enclosed by the Fram Strait,  the northern boundary of
the Barents Sea, the eastern boundary of the Kara Sea, the Bering Strait and the northern edge of the Canadian
Arctic Archipelago, shown in Figure 1.
**2.2 Reference datasets**
Uncertainty in observed variables tends to be higher in the Arctic than in many other parts of the world. There
are severe practical difficulties with collecting in situ data on a large scale over regions of ice-covered ocean.
While satellites have in many cases been able to produce Arctic-wide measurements of some characteristics,
most notably sea ice concentration, the relative lack of in situ observations against which these can be calibrated
means knowledge of the observational biases is limited. Reanalysis data over the Arctic is also more subject to
model errors than in other regions, due to errors in atmospheric forcing, and the existence of fewer direct
observations available for assimilation (Lindsay et al, 2014). The approach of this study is to use a wide range of
observational data to evaluate modelled sea ice state and surface radiative fluxes, setting results in the context of
in situ validation studies. The same datasets are then used as reference datasets for the induced surface flux
framework.
To evaluate modelled sea ice concentration, we use the HadISST1.2 dataset (Rayner et al, 2003), derived from
passive microwave observations. To evaluate modelled sea ice thickness Arctic-wide, we use the ice-ocean
model PIOMAS (Zhang and Rothrock, 2003), which is forced with the NCEP reanalysis and assimilates ice
concentration data. Laxon et al (2013) and Wang et al (2016) found PIOMAS to estimate anomalously low
winter ice thicknesses compared to satellite observations in some years. In particular, Wang et al (2016) found
PIOMAS to have a mean bias of -0.31m relative to observations from the ICESat (Ice, Cloud and land Elevation
Satellite) laser sensor. To set the PIOMAS comparison in context, we use two additional datasets to evaluate the
model over smaller regions; measurements from radar altimetry aboard the ERS satellites from 1993-2000
(Laxon et al, 2003), limited to latitudes below 82°N; and estimates compiled by Rothrock et al (2008), derived
from a multiple regression of submarine transects over the Central Arctic Ocean from 1975-2000, constrained to
be seasonally symmetric.
To evaluate modelled surface radiative fluxes across the whole Arctic Ocean, three datasets are used. Firstly, we
use the CERES-EBAF (Clouds and Earth's Radiant Energy Systems – Energy Balanced And Filled) dataset
(Loeb et al, 2009), based on direct measurements of top-of-atmosphere radiances from EOS sensors aboard
NASA satellites, available from 2000 – present. Secondly, we use the ISCCP-FD (International Satellite Cloud
Climatology Project FD-series) product (Zhang et al, 2004). Lastly, we use the ERA-Interim (ERAI)
atmospheric reanalysis dataset, which provides gridded surface flux data from 1979-present using a reanalysis
system driven by the ECMWF (European Centre for Medium-range Weather Forcecasts) IFS forecast model
and the 4D-Var data assimilation system (Dee et al, 2011).
In-situ validation of these datasets in the Arctic has been limited, but Christensen et al (2016) found CERES to
perform quite well relative to other products, albeit underestimating downwelling LW fluxes from November –
February by 10-20 $Wm^{-2}$ relative to in situ observations at Point Barrow (Alaska). Liu et al (2005) found
ISCCP-FD to simulate SW radiative fluxes fairly accurately relative to observations from SHEBA, but to
underestimate downwelling SW fluxes in spring by over 30 $Wm^{-2}$, also overestimating downwelling LW fluxes
in winter by around 40 $Wm^{-2}$ . Lindsay et al (2014) identified ERAI as producing a relatively accurate
simulation of surface fluxes compared to in situ observations at Point Barrow (Alaska) and Ny-Ålesund
(Svalbard), although tending to underestimate downwelling SW fluxes in the spring by up to 20 $Wm^{-2}$ and
overestimate downwelling LW fluxes in the winter by around 15 $Wm^{-2}$. Comparison of winter downwelling LW
fluxes in all datasets to in situ measurements compiled by Lindsay et al (1998) suggests that while ISCCP-FD is
likely to be biased high, ERAI and CERES may be relatively accurate.
In addition to the datasets above, in section 4 we make use of satellite estimates of date of melt onset over sea
ice (Anderson et al, 2012), also derived from passive microwave sensors. In section 5, to evaluate the impact of
observational uncertainty in ice area, we use the NSIDC 'Sea Ice Concentrations from Nimbus-7 SMMR and
DMSP SSM/I-SSMIS Passive Microwave Data, Version 1' (Cavalieri et al, 1996) and HadISST.2 (Titchner and
Rayner, 2014). Finally, in section 6, the CERES-SYN dataset (Rutan et al, 2015), similar to CERES-EBAF but
available at higher temporal resolution, is used to examine modelled surface radiation evolution during May in
more detail.

## 3. Evaluating HadGEM2-ES

In this section, and throughout the rest of the paper, a difference between a model simulation of a particular
quantity, and any reference dataset for that quantity, is referred to as a 'bias'. In a similar way, the difference in
model surface flux judged to arise from the difference in a particular quantity relative to a reference dataset is
referred to as an 'induced surface flux bias'. Attention is drawn to the fact that, due to observational inaccuracy,
true model bias relative to the real world may be somewhat different from the biases described in this way.
Modelled September sea ice extent in HadGEM2-ES from 1980 to 1999 is systematically lower than that
observed (Figure 2a); the four members of the HadGEM2-ES historically-forced ensemble simulate a mean
September sea ice extent of 5.78 x $10^6$ km$^2$, with ensemble standard deviation of 0.24 x $10^6$ km$^2$. By
comparison, the mean observed September sea ice extent over this period was 6.88 x $10^6$ km$^2$ according to the
HadISST1.2 dataset.
Mean ice thickness is consistently lower than that estimated by PIOMAS for the Arctic Ocean region (Figure
2b), with the highest biases of -0.4m occurring in October, close to the minimum of the annual cycle, and a
near-zero bias in May, close to the maximum.. Modelled ice thickness is also biased low relative to the ERS
satellite measurements (Figure 2c), with thickness biases ranging from -0.57m in November to -0.16m in April.
Finally, modelled ice thickness is biased low relative to the submarine data (Figure 2d), with thickness biases
ranging from  -1.5m in August to -0.8m in January and May. Hence there is clear evidence of a low model bias
in annual mean ice thickness, but there is also evidence of a model bias in the seasonal cycle of volume balance,
as model biases at minimum ice thickness tend to be larger (i.e. more negative) than those at maximum ice
thickness, implying excess ice melting and ice freezing in the model.
Relative to PIOMAS, a volume balance bias of 38cm is implied. This means the model simulates excess ice
melt in summer, and excess ice growth in winter, equivalent to 38cm on average over the area of the Arctic
Ocean region. Relative to the ERS satellite measurements, a volume balance bias of 42cm is implied. As the
seasonal cycle of submarine ice thickness is constrained to be symmetric, a more useful measure of the volume
balance bias here can be gained by comparing the maxima and minima of the submarine ice thickness seasonal
cycle to that of HadGEM2-ES; in this case, a volume balance bias of 38cm excess melt in summer, and excess
growth in winter, is implied. Hence HadGEM2-ES is likely to overestimate the magnitude of the ice thickness
seasonal cycle by around 40cm across the Arctic Ocean on average. In particular, the large negative model bias
in ice thickness in late summer is likely to be the principal cause of the low bias in September ice area.
Maps of the ice thickness bias in April and October (Figure 3b-d) show agreement that the thin ice thickness
bias is smaller on the Pacific side of the Arctic than on the Atlantic side of the Arctic, becoming very small or
positive in the Beaufort Sea. There is also striking agreement in the spatial pattern of the amplification bias of
the seasonal cycle, as diagnosed by April-October ice thickness difference (Figure 3). All three ice thickness
datasets show the HadGEM2-ES ice thickness seasonal cycle to be too amplified across much of the Arctic, by
up to 1m in the Siberian shelf seas; in addition, all show that in the Beaufort Sea, the amplification is
nonexistent or even negative. There is clear correspondence between areas where modelled annual mean ice
thickness is biased low (high), and areas where the modelled seasonal cycle is over-amplified (under-amplified).
This is likely to be associated with the ice thickness-growth feedback, whereby a steeper temperature gradient
induces stronger conduction and hence ice growth for thin ice. Indeed, negative correlations between summer
sea ice and sea ice growth the following winter are a ubiquitous feature of CMIP5 models (Massonnet et al,

35 2018).

The bias in ice volume balance is associated with a bias in ice energy uptake. In calculating this bias, and for the
rest of the study, we assume an ice density of 917 kgm$^{-3}$, the constant value used by HadGEM2-ES. For
example, the 40cm ice melt bias during summer would be associated with an energy uptake bias of around 1.5 x
$10^8$J, or 15Wm$^{-2}$ over a 4-month melting season; the 40cm ice growth bias during winter would be associated
with an energy uptake bias of -7.5Wm$^{-2}$ over an 8-month freezing season. The ice energy uptake has three
drivers: the surface energy balance, the oceanic heat convergence, and ice divergence. Sea ice divergence is
generally recognised to be a small term (e.g. Serreze et al, 2007). Although Arctic Ocean heat convergence can
be significant in size, across much of the Arctic the sea ice is insulated from the main source of heat energy
from beneath, the warm Atlantic water layer, by fresh water derived mainly from river runoff (e.g. Serreze et al,
2006; Stroeve et al, 2012b). Because of this, in the Arctic Ocean interior direct solar heating of the ocean is a
much larger contributor to sea ice basal melting than oceanic heat convergence, as observed by Maykut and
McPhee (1995), McPhee et al (2003) and Perovich et al (2008), and modelled by Steele et al (2010) and Bitz et
al (2005). In particular, it has been found that in HadGEM2-ES oceanic heat convergence is of negligible
importance to the sea ice heat budget (Keen et al, 2018). Hence for the main purposes of this study, we
concentrate on the surface energy balance, and neglect the other two terms (although the ocean heat
convergence is briefly discussed in Section 6).
Surface radiative fluxes are now evaluated. In the following discussion, and throughout this study, the
convention is that positive numbers denote a downwards flux. Fluxes of downwelling SW radiation are higher in
HadGEM2-ES than in all observational estimates during the spring (Figure 4a-c), with May biases of 22, 43 and
53 Wm$^{-2}$ relative to CERES, ERAI, and ISCCP-FD respectively.  During the summer, upwelling SW radiation
is consistently lower in magnitude than in HadGEM2-ES, with June biases of 16, 37 and 44 Wm$^{-2}$ with respect
to ERAI, CERES and ISCCP-FD respectively (a positive bias in an upward flux demonstrates that the modelled
flux is too low in magnitude). There is no consistent signal for a low bias in downwelling SW during the
summer, suggesting a negative model surface albedo bias. The effect is that modelled net downward SW flux is
too large with respect to all observational datasets in May and June, and with respect to some in July and
August. Relative to CERES, the May downwelling SW bias displays no clear spatial pattern over the Arctic
Ocean (Figure 4a), but the June upwelling SW bias, and hence the net SW bias, tend to be somewhat higher in
magnitude towards the central Arctic (Figure 4b-c).
As the June net SW bias is likely to result from a surface albedo bias we discuss the parameters affecting surface
albedo over sea ice in HadGEM2-ES: ice fraction, snow thickness and surface melt onset. Ice fraction has
already been evaluated and for snow thickness no reference dataset is available; however, surface melt onset can
be evaluated using satellite observations (Figure 6). We define the date of melt onset for any grid cell as the first
day on which the surface temperature exceeds -1°C (varying this threshold by 0.5°C in either direction changes
the date in only a small minority of grid cells). The average date of melt onset as estimated by this method
(Figure 5a) is then compared to that measured by the satellite-derived dataset described in Section 3 (Figure 5b),
with model bias shown in Figure 5c. Large spatial variability is evident in the observations. Melt onset occurs in
early to mid-May around the Arctic Ocean coasts, but much later in the Central Arctic, around mid-June. In
contrast, the HadGEM2-ES surface melt onset date is in mid- to late May across the Arctic Ocean, without the
strong gradients seen in the observations. This would cause a surface albedo, and hence net SW, bias with a
strong maximum in the Central Arctic, similar to that discussed above.
We now return to the radiation evaluation to evaluate LW fluxes. Fluxes of longwave (LW) radiation are lower
in magnitude in HadGEM2-ES throughout the winter than in all observational datasets (Figure 5d-f). For

downwelling LW, the mean model biases from December-April are -16, -22 and -40 Wm$^{-2}$ for ERAI, CERES and ISCCP-FD respectively; for upwelling LW, the biases are 11, 16 and 18 Wm$^{-2}$ for CERES, ERAI and ISCCP respectively. Because the downwelling LW biases vary more than the upwelling LW biases, there is uncertainty in inferring a model bias in net downwelling LW; ISCCP suggests a large model bias of -22 Wm$^{-2}$, CERES a smaller bias of -11 Wm$^{-2}$, while ERAI suggests a bias of only 1 Wm$^{-2}$. As in situ studies have shown both underestimation (by CERES) and overestimation (by ERAI and ISCCP-FD) of downwelling LW in winter, there is no clear indication as to where the true model bias in this quantity may lie. Maps of the downwelling and net down LW bias relative to CERES in February (Figure 5d,f) show the bias tends to be somewhat higher towards the North American side of the Arctic, and lower on the Siberian side.

The surface radiation evaluation provides clues as to the causes of the HadGEM2-ES ice volume balance bias, but also underlines why a more detailed analysis is required to properly quantify these causes. For example, in winter the low downwelling LW bias provides a clear mechanism for the bias in ice freezing. However, the reference datasets also suggest a low bias in upwelling LW that at first sight would tend to counteract this.. In fact, these biases are fully consistent: a low bias in downwelling LW would be expected to cause a low surface temperature bias, causing both a high bias in ice growth and a low bias in upwelling LW. The qualitative evaluation fails to capture the full causal relationship; for this, an analysis of exactly how the downwelling LW bias affects surface flux, including the upwelling LW response, is necessary.

In the summer, meanwhile, the surface radiation evaluation suggests that a bias in net SW radiation is responsible for the ice volume balance bias, and that this in turn is related to a surface albedo bias. However, at least two possible drivers of this have been identified: the surface melt onset bias, and the underlying ice area bias that is itself likely to be caused by the ice volume balance bias. Quantifying the extent to which each driver is important, and at which times of year, is likely to help in resolving this circular causal loop. In the next section, it is described how the effects of each model bias on the sea ice volume balance can be separated and quantified through their effect on the surface flux, in the ISF framework.

## 4. The induced surface flux (ISF) framework: a way to quantify the effect of each model bias on sea ice volume balance

We are motivated by the observation that each of the model biases described above affects the sea ice volume balance by acting through the total downwards surface energy flux (referred to as the surface flux). An excess of downwelling radiation leads directly to a higher surface flux, higher sea ice energy uptake and a bias towards ice melting. A bias in ice area, or in surface melt onset, is associated with a bias in surface albedo, hence a bias in net SW, and in the total surface flux. Finally, a bias in ice thickness alters the thermodynamics of the entire snow-ice column, and of the near-surface atmosphere layer. Flux continuity considerations show that in the freezing season, thinner ice is associated with a warmer surface temperature, and hence a higher upwelling LW flux and a smaller total downwards surface energy flux. Thinner (thicker) ice is also associated with a stronger (weaker) upwards conduction flux, and hence stronger (weaker) ice growth.

Each of these relationships can be quantified, in principle, at any point in model space and time. Specifically,
the rate at which the surface flux depends on each variable alone, with others being held constant, can be
estimated. To this end, we approximate the surface flux $F_{sfc}$ at each point in model space $x$ and time $t$ by an
explicit function $g_{x,t}$ of quasi-independent variables $v_i$. The functions $g_{x,t}$ are constructed in such a way as
to capture each of the relationships described above in a manner that best represents both HadGEM2-ES, and
also the conditions at the point $x$ and time $t$. In addition, the function captures the indirect effect of any model
bias on surface flux via surface temperature and upwelling LW, which will tend to counteract the direct effect to
a degree. Hence the dependence of the surface flux on each of the independent variables at point $x$, time $t$ can be
approximated by $\left[\partial g_{x,t}/\partial v_i\right]^{MODEL}$. Given a model bias in variable $v_i$ at $(x,t)$ we can then estimate the surface
flux bias induced by that model bias as $\left[\partial g_{x,t}/\partial v_i\right]^{MODEL} \partial g_{x,t}/\partial v_i \left(v_{i,x,t}^{MODEL} - v_{i,x,t}^{REFERENCE}\right)$.
The function $g_{x,t}$ and its derivation are described fully in Appendix A, but are summarised briefly here. The
surface flux is expressed as a sum of separate radiative and turbulent components. Ice-covered and ice-free
portions of a grid cell are treated separately; in ice-covered areas, dependence on surface temperature is
linearised, and flux continuity at the surface and a uniform vertical conductive flux through the ice are assumed,
allowing surface temperature to be eliminated, and the dependence of surface flux on upwelling LW to be
captured. The surface albedo, upon which the upwelling SW component depends, is expressed in terms of ice
fraction, snow fraction and melt onset occurrence, based on the albedo parameterisation of HadGEM2-ES.
Latent heat flux over ice is neglected. In this way the surface flux is expressed as an explicit function of
downwelling shortwave (SW) and longwave (LW) radiation, ice concentration, category ice thickness, snow
thickness, sensible and latent heat fluxes over ice and ocean, and melt onset occurrence (a logical determining
whether or not the snow surface is undergoing melting). Induced surface flux due to ice thickness bias is
determined by summing each of the separate ISF biases by category.
The usefulness of this approach is that surface flux operates linearly on the sea ice volume balance, meaning that
each of the ISF biases at $(x,t)$ can be averaged over large regions of time and space to understand large-scale sea
ice biases. Clearly, none of the driving model biases operate on the sea ice state in a linear sense. For example,
given identical surface melt onset (and hence surface albedo) biases at different points in the Arctic, each could
have very different implications for local sea ice volume balance, depending on the downwelling SW modelled
at each point. Conversely, identical downwelling SW biases would have different implications for sea ice
volume balance depending on the modelled surface albedo. Model bias in ice thickness behaves in a particularly
nonlinear fashion, with bias in regions of thinner ice having far more influence than that in regions of thicker
ice. The effect of estimating ISF bias at each point separately, and then averaging to determine large-scale
effects, is to bypass all nonlinearities.
A second advantage of this approach lies in the quasi-independence of the variables: while each variable may
affect the others over timescales varying from days to months, each affects the surface flux instantaneously (in
HadGEM2-ES). Hence a model bias in any variable represents an effect on the surface flux that is separable
from the effect of a model bias in any other. If the surface flux variation is completely described by the function
$g_{x,t}$ , therefore, the sum of the ISF biases, over all variables, must approach the true model surface flux bias
(although it will be seen that this claim is impossible to evaluate precisely due to observational uncertainty). In
this way, large-scale model biases in surface flux, and hence sea ice volume balance, can be broken down into
separate contributions from model biases in each of the independent variables.
The ISF calculation process is now illustrated for two processes in turn. The model bias in melting surface
fraction for the month of June 1980 is positive over most of the Arctic, although only weakly so towards the
coasts (Figure 6a), reflecting melt onset modelled earlier than observed during this month. The reference dataset
is derived from SSMI microwave observations. The rate of change of surface flux with respect to melt onset
occurrence tends to be higher in the Central Arctic (Figure 6b). This reflects a greater tendency to clear skies in
the Central Arctic, as this field is effectively downwelling SW multiplied by the difference in parameterised
albedos. The product of these two fields represents the modelled surface flux bias induced by the model bias in
melt onset; this is also positive over most of the Arctic Ocean, by up to 25 Wm$^{-2}$ in the central Arctic, reflecting
the greater absorption of SW radiation induced by the early melt onset (Figure 6c)
The model bias in downwelling LW radiation in February 1980 is predominantly negative (Figure 6d); here
CERES-EBAF is the reference dataset. The rate of dependence of surface flux on downwelling LW (Figure 6e)
is everywhere between 0 and 1, tending to be lower in regions of thicker ice, associated with a greater tendency
for downwelling LW biases to be counteracted by upwelling LW biases in these regions. The resulting ISF bias
(Figure 6f) is negative almost everywhere, but is lower in magnitude than the driving downwelling LW bias.
**5. Induced surface flux biases**
**5.1 Aggregate ISF biases**
We calculate surface flux biases induced by model biases in downwelling SW, downwelling LW, ice area, local
ice thickness and surface melt occurrence (the variables for which reference datasets are available). The
resulting fields are averaged over the model period and over the Arctic Ocean region, to produce for each
variable a seasonal cycle of total surface flux bias induced by the bias in that variable. The induced surface flux
(ISF) biases are displayed in Figure 7, together with total ISF bias, net radiative flux bias estimated by the direct
radiation evaluation relative to ISCCP-FD, CERES and ERAI, and also sea ice energy uptake biases implied by
the seasonal ice volume balance bias relative to PIOMAS. The ISF biases are also shown in Table 1, using
CERES as reference dataset for the radiative terms. During winter, ISF biases generally sum to negative values,
indicating that model biases in this season induce net additional surface energy loss and ice growth. During
summer, ISF biases generally sum to positive values, indicating additional net surface energy gain and ice melt.
In both seasons, these results are consistent with the radiation and ice volume balance evaluation.
Major roles are identified for particular processes in certain months. Firstly, in June the surface melt onset bias,
through its effect on the surface albedo, induces a surface flux bias of -13.6 Wm$^{-2}$, equivalent roughly to an
extra 11cm of melt. Secondly, in August a bias in ice fraction induces a surface flux bias of 9.6 Wm$^{-2}$,

equivalent to an extra 8cm of melt. This is associated with the overly fast retreat of sea ice in HadGEM2-ES, and the low extents in late summer, as noted in Section 3. Thirdly, from October-March the downwelling LW biases induce substantial surface flux biases ranging from -6.5 to -3.8 $Wm^{-2}$ depending on choice of reference dataset, equivalent to additional sea ice growth of 20-33cm. Finally, from November-March the negative ice thickness bias induces substantial surface flux biases reducing from -8.3 $Wm^{-2}$ to -2.0 $Wm^{-2}$ as the freezing season progresses, equivalent to total additional ice growth of 24cm.

Internal variability in the ISF biases is measured by taking the standard deviation of the whole-Arctic ISF bias for each process and month across all 20 years in the model period, and all four ensemble members used. Variability is highest in the ice area term, reaching 4.0 $Wm^{-2}$ in July. Variability reaches considerable size in some other terms in some months, for example 1.1 $Wm^{-2}$ for surface melt onset in June, 1.9 $Wm^{-2}$ for ice thickness in November, but is otherwise mainly under 1 $Wm^{-2}$ in magnitude. In each case, therefore, the ISF biases noted above are persistent features of the model; surface melt onset and ice fraction biases induce additional ice melt in summer, while downwelling LW and ice thickness biases induce additional ice growth in winter. The total summer volume balance bias accounted for by the analysed processes is of the order 20cm, while the total winter volume balance bias is of the order 45-55cm depending on radiation reference dataset.

## 5.2 Spatial variability

Examining first the June melt onset ISF bias, the spatial pattern of the bias is characterised by a maximum in the central Arctic, with values falling away towards the coast; this is directly related to the spatial pattern of the melt onset bias itself shown in Figure 6. It is very similar to the spatial pattern of the directly evaluated net SW bias in this month, providing additional evidence that the melt onset bias is the principal cause of this. This implies that the additional ice thinning induced by this bias is greatest in the Central Arctic and least at the coasts.

The August ice concentration ISF bias displays a sharply defined pattern, with high values across the shelf seas and the Atlantic side of the Arctic falling to low or negative values in the Beaufort Sea, again very similar to the pattern of the ice concentration bias itself. The implication is that the model bias towards ice thinning in August is largely based in areas where ice concentration is already biased low.

The November-March ice thickness ISF bias displays a pattern which is almost identical to that of the August ice fraction ISF bias, but with the opposite sign, with high negative values on the Atlantic side of the Arctic rising to near-zero values in the Beaufort Sea. Hence this model bias has the reverse effect to that of the ice concentration bias, reducing existing ice thickness biases by promoting additional ice growth in these areas.

Finally, the winter downwelling LW ISF bias is much more spatially uniform, but displays slightly higher values on the Pacific side of the Arctic than the Atlantic side, a different pattern to that displayed by the downwelling LW itself in Figure 4d. The contrast is due to the role the effective ice thickness scale factor plays in determining the induced surface flux bias; the thicker ice present on the American side of the Arctic, tends to greatly reduce the flux bias. This represents the thickness-growth feedback; thicker ice will grow less quickly

than thin ice under the same atmospheric conditions. The downwelling LW bias tends to increase ice growth Arctic-wide, but less so in regions where ice is already thick.

The spatial patterns of total ISF bias shows many similarities to the total net radiation bias evaluated by CERES in most months of the year (Figure 8), notably a tendency in July and August for positive surface flux biases to be concentrated on the Atlantic side of the Arctic, and a tendency throughout the freezing season for negative surface flux biases to be least pronounced in the Beaufort Sea, where the ice thickness biases are lowest. We note that the spatial pattern of amplification of the ice thickness seasonal cycle displayed in Figure 3 is very similar, with amplification most pronounced in the Atlantic sector, and least pronounced in the Beaufort Sea. The surface flux biases produced by ice fraction biases in August, and ice thickness biases in November, provide reasons for the spatial variation in amplification of the ice thickness seasonal cycle seen in Figure 4, as well as the close resemblance of this pattern to the model biases in annual mean ice thickness. Ice which is thinner in the annual mean will tend to melt faster in summer, due to the net SW biases associated with greater creation of open water (the surface albedo feedback), and to freeze faster in winter, due to greater conduction of energy through the ice (the ice thickness-growth feedback).

### 5.3 Using the ISF biases to separate sea ice forcings and feedbacks

It is helpful to divide the processes examined into feedbacks (surface flux biases induced by biases in the sea ice state itself) and forcings (those induced by variables external to the sea ice state). In this sense, a 'forcing' refers to a variable which is independent of the sea ice volume on instantaneous timescales, rather than being used in the traditional sense of a radiative forcing. Of the variables examined, downwelling SW and LW radiation, as well as the surface melt onset, have this property, and hence their corresponding ISF biases can each be regarded as a 'forcing' on the sea ice state. However, ice thickness and area do not have this property, and their corresponding ISF biases should be regarded instead as intrinsic feedbacks of the sea ice state.

The ice concentration ISF flux bias (specifically during the melting season) can be identified with the effect of the surface albedo feedback on the sea ice state. During the melting season the ice area affects the estimated surface flux only through the surface albedo, and the surface flux biases induced in this way cause associated biases in ice melt.

On the other hand, the ice thickness ISF bias (specifically during the freezing season) can be identified with the effect of the thickness-growth feedback on the sea ice state. This is perhaps less obvious, as the ice thickness affects the estimated surface flux via the surface temperature and upwelling LW radiation, while the thickness-growth feedback is usually understood to result from differences in conduction. However, the assumption of flux continuity at the surface in constructing the estimated surface flux means that the cooler surface temperatures, and shallower temperatures gradients occurring for thicker ice categories are manifestations of the same process. Slower ice growth at higher ice thicknesses is caused by a smaller negative surface flux, and the surface temperature is the mechanism by which this is demonstrated. Hence the effect of the thickness-growth feedback is described by the ice thickness-induced component of the surface flux bias.

The ISF analysis allows the effect of the surface albedo and thickness-growth feedbacks on the sea ice state to be quantified, and compared to the effect of external drivers. Arctic-wide, the surface albedo feedback, diagnosed as the ice area-induced component of the surface flux bias, contributes an average of 5.2 Wm$^{-2}$ to the surface flux bias over the summer months, equivalent to an extra 13cm of ice melt. This is very similar to the effect of the surface melt onset-induced component, which contributes an average of 5.3 Wm$^{-2}$, equivalent also to an extra 13cm of ice melt. In the freezing season, meanwhile, the thickness-growth feedback, diagnosed as the ice thickness-induced component of the surface flux bias, contributes an average of -4.4 Wm$^{-2}$ to the surface flux bias from October-April, equivalent to an extra 26cm of ice freezing, while the downwelling LW-induced component (using CERES as reference dataset) contributes an average of -4.9 Wm$^{-2}$, equivalent to an extra 29cm of freezing over this period.

### 5.4 ISF residuals and observational uncertainty

The ISF biases, summed over all independent variables, should approach the true total surface flux bias. However, this is difficult to evaluate as the true surface flux bias is not known. Hence it is necessary to use proxy quantities to evaluate the total ISF bias: directly evaluated surface net radiation bias (relative to ISCCP-FD, ERAI and CERES respectively); and ice energy uptake bias, derived from ice volume balance bias relative to PIOMAS.

For most months of the year, all estimates of total ISF bias fall within the spread of these four datasets (Figure 6), the exceptions being June and July when total ISF bias is smaller than all surface flux proxies. However, the spread is extremely large. For example, in the month of January the estimates of total ISF bias are -12.3, -8.2 and -6.1 Wm$^{-2}$ (with ISCCP-FD, CERES and ERAI used as downwelling radiation datasets respectively), while the estimates of net radiation bias are -18.2, -11.6 and 0.6 Wm$^{-2}$ from ISCCP-FD, CERES and ERAI respectively, and ice heat uptake bias is estimated as -10.1 Wm$^{-2}$. Hence it is difficult to evaluate the total ISF bias within current observational constraints, and at best it can be said that the total ISF bias is qualitatively consistent, over the year as a whole, with the surface flux bias proxies. A possible cause of the lower total ISF bias in June and July is the 'missing process' of snow on ice, which cannot be evaluated here due to the lack of a reference dataset. The early surface melt onset, and sea ice fraction loss, as modelled by HadGEM2-ES, would be associated with an early loss of snow on ice, with an additional surface albedo bias and hence an additional ISF bias.

On the other hand, the annual mean ice heat uptake bias (0.0 Wm$^{-2}$) provides a strong constraint on the annual mean surface heat flux bias, in the absence of a significant oceanic heat convergence contribution. For example, the annual mean total ISF biases are -3.6 Wm$^{-2}$ and -4.5 Wm$^{-2}$ when CERES and ISCCP-FD are used as reference datasets respectively; these would imply sea ice thickening of 7m and 9m over the 1980-1999 in HadGEM2-ES, which does not occur. Hence in the annual mean, the total ISF bias is too low. This annual mean bias is related to the tendency for the ISF analysis to account for a greater bias towards ice growth in winter (45-55cm), than that towards ice melt in summer (20cm). It is likely to derive, at least in part, from the use of

multiple reference datasets whose errors are not constrained to correlate in a physically realistic sense, but may also be related to the missing processes in June and July.

Observational error is one potential cause of error in the ISF biases. An idea as to the potential magnitude of this can be seen from the large spread in SW and LW ISF bias (across different datasets) during summer (Figure 6). For example, in July the model downwelling LW bias with respect to ERAI produces an aggregated ISF bias of -7.0 $Wm^{-2}$, but that with respect to CERES produces an aggregate ISF bias of 8.0 $Wm^{-2}$. Calculation of ice area ISF biases using NSIDC and HadISST.2 as reference, described in section 2.2 and not shown here, showed a similar magnitude of uncertainty in the ice area term ($\pm10Wm^{-2}$ in summer, $\pm2Wm^{-2}$ in winter). We note that the evidence from in situ validation studies suggests that the winter downwelling LW estimates of ERAI and CERES are more likely to be accurate than that of ISCCP-FD. Hence the downwelling LW ISF bias is likely to be estimated more accurately when ERAI or CERES are reference dataset, and the bias towards ice growth is likely to lie closer to the lower end of the range (20cm).

In Appendix B, inherent theoretical errors in the ISF analysis are discussed and are found to be small relative to the sensitivity to use of observational datasets. The largest errors are listed here: firstly, due to sub-monthly variation in the component variables, the winter downwelling LW component may be underestimated in magnitude by around 0.6 $Wm^{-2}$ on average, and the ice area component in August may be overestimated by around 1.6$Wm^{-2}$. Secondly, due to the evaluation of surface flux dependence at a model state which is itself biased, the total ISF bias in October is overestimated in magnitude by around 3.6 $Wm^{-2}$. Thirdly, due to nonlinearities in the surface flux dependence on ice thickness, the ice thickness component is overestimated in magnitude by 0.7 $Wm^{-2}$ on average from October-April, with a maximum overestimation in November of 1.9 $Wm^{-2}$. As these biases are, in the main, considerably smaller than the differences between ISF biases when different reference datasets are used, it is concluded that observational errors are the more important contribution to error in the ISF biases.

**6.   Discussion**

**6.1 Using the ISF framework to understand the HadGEM2-ES sea ice state**

The HadGEM2-ES ISF biases are qualitatively consistent with the direct net radiation evaluation and with the sea ice simulation, both in terms of seasonal and spatial variation, and allow the effect of the surface albedo feedback and thickness-growth feedback on the sea ice volume balance to be separated. This allows the HadGEM2-ES sea ice biases to be understood by considering, in turn, the separate ISF components, their magnitudes, and the times of year when they are important. The anomalous summer sea ice melt is initiated by the early melt onset occurrence, and maintained by the surface albedo feedback, which acts preferentially in areas of thinner ice. The anomalous winter ice growth is maintained both by the thickness-growth feedback (occurring mainly in areas of thinner ice, of greater importance in early winter) and by the downwelling LW bias (more spatially uniform, in late winter). It is unclear that any significant role is played by the downwelling SW bias, as at the only time of year when the radiation datasets agree that this bias is of significant value (May),

the induced surface flux biasis more than balanced by that induced by downwelling LW. However this may
have a role in causing the later melt onset bias, as discussed below.
The means by which the external forcings – anomalous LW winter cooling, and early late spring melt onset –
cause an amplified seasonal cycle in sea ice thickness are clear. It can also be seen how, in the absence of other
forcings, these combine to create an annual mean sea ice thickness that is biased low, as seen in Section 4. The
melt onset forcing, by inducing additional ice melting through its effect on the ice albedo, enhances subsequent
sea ice melt through the surface albedo feedback. The downwelling LW bias, on the other hand, by inducing
additional ice freezing through its cooling effect, attenuates subsequent sea ice freezing through the thickness-
growth feedback. Surface flux biases induced by melt onset occurrence are enhanced, while those induced by
downwelling LW are diminished.
Acting together, the ice thickness-growth feedback and surface albedo feedback create a strong association
between lower ice thicknesses and amplified seasonal cycles, because ice which tends to be thinner will both
grow faster during the winter, and melt faster during the summer. Hence the melt onset bias, acting alone, would
induce a seasonal cycle of sea ice thickness lower in the annual mean, but also more amplified, than that
observed, because the surface albedo and thickness-growth feedbacks act to translate lower ice thicknesses into
faster melt and growth. For similar reasons, the downwelling LW bias, acting alone, would induce a seasonal
cycle of sea ice thickness higher in the annual mean, and also less amplified, than that observed. The bias seen
in HadGEM2-ES is a result of the melt onset bias 'winning out' over the downwelling LW, due to its occurring
at a time of year when the intrinsic sea ice feedbacks render the ice far more sensitive to surface radiation. The
anomalously low ice cover in September arises as a consequence of the low annual mean ice thickness, and in
particular of the anomalously severe summer ice melt. The finding that the low annual mean ice thickness is
driven by surface albedo biases is consistent with the finding by Holland et al (2010) that variance in mean sea
ice volume in the CMIP3 ensemble was mostly explained by variation in summer absorbed SW radiation.
The feedbacks of the sea ice state explain the association between spatial patterns of annual mean ice thickness
bias and ice thickness seasonal cycle amplification. However, the external forcings (melt onset and downwelling
LW bias) cannot entirely explain the spatial patterns in the mean sea ice state biases, because on a regional scale
effects of sea ice convergence, and hence dynamics, become more important. The annual mean ice thickness
bias seen in HadGEM2-ES is associated with a thickness maximum on the Pacific side of the Arctic, at variance
with observations which show a similar maximum on the Atlantic side. It was shown by Tsamados et al (2013)
that such a bias could be reduced by introducing a more realistic sea ice rheology.
**6.2 Looking beyond proximate drivers**
As noted in the Introduction, the ISF framework can only identify the proximate causes of sea ice biases. Here,
we briefly discuss causes of the two external drivers identified as causing sea ice model biases. Underestimation
of wintertime downwelling LW fluxes in the Arctic is known to be a widespread model bias in the CMIP5
ensemble (e.g. Boeke and Taylor, 2016). Pithan et al (2014) showed that this bias was likely to be a result of
insufficient liquid water content of clouds forming in subzero air masses, resulting in a failure to simulate a
particular mode of Arctic winter climate over sea ice; the 'mild mode', characterised by mild surface

temperatures and weak inversions, whose key diagnostic is observed to be a net LW flux of close to 0 Wm$^{-2}$ (Stramler et al, 2011; Raddatz et al, 2015 amongst others). HadGEM2-ES was not one of the models assessed by Pithan et al (2014), but its winter climate simulation displays many of the characteristic biases displayed by the other models, notably a tendency to model very low cloud liquid water fractions during winter compared to MODIS observations (Figure 9a) and a failure to simulate the milder mode of Arctic winter climate as demonstrated in SHEBA observations, diagnosed by 6-hourly fluxes of net LW (Figure 9b). Here we conclude that a similar mechanism is likely to be at work in HadGEM2-ES, and that insufficient cloud liquid water is the principal driver of the anomalously low downwelling LW fluxes.

The causes of the early melt onset bias of HadGEM2-ES are harder to determine. For most of the spring, comparison of daily upwelling LW fields of HadGEM2-ES to CERES-SYN observations (not shown) shows the Arctic surface to be anomalously cold in the model, as during the winter. During May, however, upwelling LW values rise much more steeply in the model, and surface melt onset commences during mid-to-late May, far earlier than in the satellite observations. A possible cause of the overly rapid surface warming during May is the zero-layer thermodynamics approximation used by HadGEM2-ES, in which the ice heat capacity is ignored. Comparing fields of surface temperature in HadGEM2-ES between the beginning and the end of May shows a 'missing' ice sensible heat uptake flux of 10-30 Wm$^{-2}$ over much of the central Arctic, which would in turn be associated with a reduction of flux into the upper ice surface of 5-15 Wm$^{-2}$. Examination of modelled and observed daily time series of downwelling LW and net SW fluxes in late May and early June suggests that a surface flux reduction of this magnitude could delay surface melt by up to 2 weeks, a substantial part of the modelled melt onset bias seen.

Another cause of the rapid warming may be the increasing relative magnitude of the downwelling SW response to cloud biases, as May progresses (compared to the downwelling LW response). Comparison of 5-daily means of HadGEM2-ES radiative fluxes during May to those from the CERES-SYN product (not shown) support this hypothesis; a modelled bias in downwelling SW grows quickly during early May, from ~ 0 Wm$^{-2}$ to ~ 30 Wm$^{-2}$, while the modelled bias in downwelling LW remains roughly constant.

**6.3 Missing processes in the ISF analysis**

The ISF analysis, as presented, does not comprise an exhaustive list of processes affecting Arctic Ocean surface fluxes. The missing process of the effect of snow fraction on surface albedo has already been noted, and its likely effect on the total June and July ISF bias. We note also that the direct effect of thinning ice on ice albedo could induce an additional flux bias relative to the real world, despite the fact that this effect is not represented in HadGEM2-ES. The effect of snow thickness bias on winter conduction and surface temperature is another such process which cannot be included due to inadequate observations. Model biases in the turbulent fluxes may also be significant. While the process which is likely most important in determining these during the winter is captured (ice fraction in the freezing season), a more detailed treatment of turbulent fluxes would also examine the effect on these of the overlying atmospheric conditions. It is also noted that snowfall itself is a component of the surface flux that could, in theory, be evaluated directly given a sufficiently reliable observational reference.

A complete treatment of model biases affecting the sea ice volume budget would also examine causes of bias in oceanic heat convergence. For the reasons discussed in Section 4, these are likely to be small in the Arctic Ocean interior in HadGEM2-ES and observations, but the model bias could nevertheless conceivably be of considerable size in the context of the surface flux biases shown in Figure 6. The total Arctic Ocean heat convergence modelled by HadGEM2-ES for the period 1980-1999 is 4.4 $Wm^{-2}$, although this figure shows high sensitivity to the location of the boundary in the Atlantic sector, suggesting that most of this heat is released close to the Atlantic ice edge. This figure is slightly higher than the 3 $Wm^{-2}$ found by Serreze et al (2007) in their analysis of the Arctic Ocean heat budget, but is broadly consistent with observational estimates of oceanic heat transport through the Fram Strait from 1997 to 2000 by Schauer et al, 2004 (likely to be the major contributor to Arctic Ocean heat convergence). This suggests that errors in oceanic heat convergence are unlikely to contribute significantly to sea ice volume biases in HadGEM2-ES. However, for a hypothetical model that simulated greater oceanic heat convergence in the Arctic Ocean interior, the surface flux analysis presented here would fail to adequately describe the model bias in the sea ice volume budget.

Finally, the assumptions underpinning the ISF framework include an instantaneous effect of the independent variables on the surface flux. In the real world, and in many models, there is a time lag associated with the effect of the ice and snow thickness, due to the thermal inertia of the snow and ice. In the time taken for a change of ice thickness to cause a change in surface flux, other variables such as downwelling radiation could in theory be affected, raising doubts as to whether ice and snow thickness are truly independent variables. However, it is likely that any mechanism by which ice thickness could affect another variable would act first through the surface flux, and hence that the timescale on which ice thickness affects surface flux is shorter than that on which it affects other variables. It is also seen in this study that the ISF biases are largest, by far, in the thinnest ice category, where the effect of ice thickness on surface flux would be near-instantaneous. ISF biases in the thickest ice category, where the time lag could be of significant size, tend to be negligible.

## 7. Conclusions

A framework has been designed (the ISF framework) that allows the proximate causes of biases in sea ice volume balance to be separated and quantified. Given reference datasets for independent variables, fields of induced surface flux bias can be calculated from the underlying model bias; these in theory sum to the total surface flux bias. In practice, the total ISF bias matches both the net radiation bias, and the ice volume balance bias to first order: processes evaluated cause around 40cm additional ice growth during the ice freezing season, and 20cm additional ice melt in winter; a missing process for which we have no reference (snow thickness) is likely to account for at least some additional ice melt in summer. However, observational uncertainty in the evaluated terms prevents direct evaluation of the total ISF bias, and is the largest contribution to ISF uncertainty.

The ISF analysis enables model biases in sea ice growth and melt rate to be attributed in detail to different causes. In particular, the roles played by the sea ice albedo feedback, by the sea ice thickness-growth feedback, and by external forcings, can be quantified. The analysis reveals how the melt onset bias of HadGEM2-ES tends to make model ice thickness both low in the annual mean, and too amplified in the seasonal cycle, with the

downwelling LW bias acting to mitigate both effects. The result is consistent with the prediction of DeWeaver et al (2008) that sea ice state is more sensitive to surface forcing during the ice melt season than during the ice freeze season. The analysis also suggests that through an indirect effect on surface albedo at a time when sea ice is particularly sensitive to surface radiation biases, the zero-layer approximation, which was until recently commonplace in coupled models, may be of first-order importance in the sea ice state bias of HadGEM2-ES.

The ISF analysis also allows more detailed analysis of the spatial patterns in sea ice volume balance simulation. In particular, the mechanisms behind the near-identical spatial pattern of biases in annual mean ice thickness (likely driven by ice dynamics) and that of biases in the ice volume balance are explicitly demonstrated. Where ice thickness is biased low in the annual mean, an enhanced seasonal cycle is apparent. This is due to the ice thickness ISF bias (in freezing season) and the ice area ISF bias (in melting season), corresponding to the thickness-growth and ice albedo feedbacks. The downwelling LW and melt onset biases, by contrast, are more spatially uniform, and do not contribute to the annual mean ice thickness control on the ice volume balance.

The finding that observational uncertainty is the most important cause of uncertainty in the ISF bias calculation itself suggests that if observational uncertainty could be reduced, the ISF analysis could become a very powerful tool for Arctic sea ice evaluation. In particular, large observational uncertainties for snow cover and summer surface radiation limit the overall accuracy of the methodology presented here. The addition of freezing season snow thickness, and melt season snow fraction, would represent useful extensions to the analysis presented. An additional caveat is that the ISF framework does not consider factors influencing turbulent fluxes (with the exception of the ice area, but this contribution is subject to particularly high uncertainty). It also does not consider the influence of oceanic heat convergence on sea ice state; in HadGEM2-ES the latter is small (~10%), but might be more significant in other models.

The ISF analysis as presented here is designed specifically to approximate HadGEM2-ES, but could in principle be generalised to other models, particularly by altering the surface albedo parameterisation used here, or by using different sea ice thickness categories. The zero-layer thermodynamic assumption used in the ISF analysis is likely to be appropriate for any model during the ice freezing season, as the largest ISF biases tend to arise from the thinnest ice categories, where the zero-layer approximation is closest to reality. However, there is a question as to whether the zero-layer approximation conceals significant surface flux bias relating to ice sensible heat uptake in the late spring.

In the case study presented here, the analysis provides mechanisms behind a model bias in sea ice simulation. However, the analysis could also be used to investigate a sea ice simulation that was ostensibly more consistent with observations, to determine whether or not the correct simulation was the consequence of model biases that cause opposite errors in the surface energy budget; a negative result would greatly increase confidence in the future projections of such a model. The analysis could be also used to investigate a model ensemble, to attribute spread in modelled sea ice state to spread in the underlying processes affecting the SEB, focussing attention on ways in which spread in modelled sea ice could be reduced. It is noteworthy that Shu et al (2015) found the CMIP5 ensemble mean Arctic sea ice volume to be biased low in the annual mean, and over-amplified in the seasonal cycle, relative to PIOMAS (albeit over the entire Northern Hemisphere), suggesting that the behaviour exhibited by HadGEM2-ES may be quite common in this ensemble.

Finally, it is suggested that the ISF framework, as well as being used to compare a model to observations, could
also be used to understand the reasons for the biases of one model with respect to another. Such a comparison
would avoid the issues of observational uncertainty discussed above, enabling the contributions of the different
model variables to the surface flux biases to be evaluated more accurately. However, the choice as to which
model parameters on which to base the ISF framework would be subjective.
**Appendix A: Description and derivation of the surface flux formula used in the ISF calculation**
The ISF framework depends upon the construction, at each point in model space and time, of an explicit
function $g_{x,t}$ which approximates the surface flux as a function of quasi-independent variables $v_i$. The
functions $g_{x,t}$ are constructed as follows. We start from the standard equation for surface flux:
$$F_{sfc} = \left(1 - \alpha_{sfc} F_{SW}\right) + F_{LW\downarrow} - \varepsilon_{sfc}\sigma T_{sfc}^4 + F_{sens} + F_{lat} + F_{snowfall} \qquad \text{(A1)}$$
where the surface flux is expressed as the sum of separate radiative and turbulent components. In this equation,
$\alpha_{sfc}$ represents surface albedo, $F_{SW\downarrow}$ downwelling SW flux, $F_{LW\downarrow}$ downwelling LW flux, $\varepsilon_{sfc} = 0.98$ ice
emissivity (as parameterised in HadGEM2-ES), $\sigma = 5.67 \times 10^{-8} Wm^{-2}K^{-4}$ the Stefan-Boltzmann constant,
$T_{sfc}$ surface temperature, $F_{sens}$ sensible heat flux, $F_{lat}$ latent heat flux, and $F_{snowfall}$ heat flux represented by
the transfer of negative enthalpy from the atmosphere to the ice associated with snowfall.
Given a model grid cell $x$, over a model month $t$, the cell is classified as freezing or melting depending upon
whether the monthly mean surface temperature is greater or lower than -2°C. In the case that the cell is
classified as melting, (A1) is simplified in the following way: $T_{sfc} = T_f$, where $T_f = 0°C$, and $F_{snowfall}$ and
$F_{lat}$ are both neglected. In addition, we expand
$$\alpha_{sfc} = a_{ice}\alpha_{ice} + \left(1 - a_{ice}\right)\alpha_{ocn} \qquad \text{(A2)}$$
where $a_{ice}$ is ice concentration, $\alpha_{ice}$ mean surface albedo over sea ice and $\alpha_{ocn} = 0.06$ the albedo of open
water used by HadGEM2-ES. Finally the ice albedo is further expanded
$$\alpha_{ice} = \left(\alpha_{melt\_ice} - \alpha_{sea}\right) + I_{snow}\left(\alpha_{melt\_snow} - \alpha_{melt\_ice}\right)$$
$$+ \left(1 - \gamma_{melt}\right)\left(1 - I_{snow}\right)\left(\alpha_{cold\_ice} - \alpha_{melt\_ice}\right) + \left(1 - \gamma_{melt}\right)I_{snow}\left(\alpha_{cold\_snow} - \alpha_{melt\_snow}\right)$$

$$\text{(A3)}$$
Here $\alpha_{sea} = 0.06$, $\alpha_{melt\_ice} = 0.535$, $\alpha_{melt\_snow} = 0.65$, $\alpha_{cold\_ice} = 0.61$ and $\alpha_{cold\_snow} = 0.8$ denote
the parameterised albedos of open water, melting ice, melting snow, cold ice and cold snow respectively, and
$\gamma_{melt}$ denotes melting surface fraction as a fraction of ice area, while $I_{snow}$ is an indicator for the presence of
snow that is set to 1 or 0 depending on whether monthly mean snow thickness exceeds 1mm. Equation (A3)
mimics the parameterisation of ice albedo in HadGEM2-ES, in which the albedo of both snow and ice is
progressively reduced from the 'cold' to the 'melting' values as surface temperature rises from -1°C to 0°C.
If the grid cell is classified as freezing, we likewise ignore $F_{snowfall}$ . We assume $F_{SW\downarrow}$ and $F_{LW\downarrow}$ to be constant
across a grid cell, and parameterise $\alpha_{sfc}$ as above. The remaining terms ( $-\varepsilon_{ice}\sigma T_{sfc}^4$ , $F_{sens}$ and $F_{lat}$ ) we expect
to vary over the six different surface types present in a grid cell: open water, and the five different ice thickness
categories. Hence we express each as a sum over the surface types:
$$-\varepsilon_{ice}\sigma T_{sfc}^4 = -(1-a_{ice})\varepsilon_{ice}\sigma T_{sfc-water}^4 - \sum_{cat=1}^{5} a_{ice-cat}\varepsilon_{ice}T_{sfc-cat}^4 \qquad\text{(A4)}$$
$$F_{sens} = (1-a_{ice})F_{sens-water} + \sum_{cat=1}^{5} a_{ice-cat}F_{sens-cat} \qquad\text{(A5)}$$
$$F_{lat} = (1-a_{ice})F_{lat-water} + \sum_{cat=1}^{5} a_{ice-cat}F_{lat-cat} \qquad\text{(A6)}$$
We make the following further approximations: firstly, that $F_{lat-cat} = 0$ for all ice categories; secondly, that
$F_{sens-cat} = F_{sens-ice}$ for all categories (i.e. that the sensible heat flux does not vary across categories); thirdly,
that $T_{sfc-water} = 1.8°C$ . Lastly, for each ice category we approximate $T_{sfc-cat}^4 = A + B(T_{sfc-cat} - T_{sfc-REF})$ ,
where $A = T_{sfc-REF}^4$ and $B = 4\varepsilon_{ice}\sigma T_{sfc-REF}^3$ , $T_{sfc-REF}$ being the monthly mean surface average temperature
of the grid cell $x$.
Flux continuity implies that over each category the surface flux is equal to $F_{condtop-cat}$ , the downwards
conductive flux from the ice surface, unless surface melting is taking place; as the freezing case is being
discussed, melting is assumed to be zero. We also make the zero-layer approximation used in HadGEM2-ES,
that the sea ice has no sensible heat capacity and that conduction is therefore uniform in the vertical for each
category. This implies that
$$F_{condtop-cat} = (T_{sfc-cat} - T_{bot})R_{ice-cat} \qquad\text{(A7)}$$
where $T_{bot} = -1.8°C$ is ice base temperature and $R_{ice-cat} = \left(\dfrac{h_{ice-cat}}{k_{ice}} - \dfrac{h_{snow}}{k_{snow}}\right)$ , $k_{ice}$ and $k_{snow}$ being ice
and snow conductivity respectively, and $h_{ice-cat}$ and $h_{snow}$ local ice and snow thickness (the latter of which is
assumed to be uniform across ice categories). For each category, setting $F_{sfc-cat} = F_{condtop-cat}$ allows $T_{sfc-cat}$
to be eliminated. This results in the following equation for $F_{sfc}$ :
$$F_{sfc} \approx g_{x,t}^{w} = a_{ice}\left(F_{atmos-ice} + BT_{ocn}\right)\sum_{cat}\gamma_{ice-REF}^{cat}\left(1 - BR_{ice}^{cat}\right)^{-1} + \left(1 - a_{ice}\right)F_{atmos-ocean} \qquad \text{(A8)}$$
where
$$F_{atmos-ice} = F_{LW\downarrow} - \varepsilon_{ice}\sigma T_{sfc-REF}^{4} + F_{sens-ice} + \left(1 - \alpha_{ice}\right)F_{SW\downarrow} \qquad \text{(A9)}$$
and
$$F_{atmos-ocean} = F_{LW\downarrow} - \varepsilon_{ocn}\sigma T_{ocn}^{4} + F_{sens-water} + F_{lat-water} + \left(1 - \alpha_{ocn}\right)F_{SW\downarrow} \qquad \text{(A10)}$$
Hence we approximate the surface flux as a function of ice area, category ice thickness, snow thickness,
downwelling SW, downwelling LW, melting surface fraction, sensible heat fluxes over ice and open water, and
latent heat flux over open water. In the ISF analysis, we analyse the resulting dependence of surface flux on ice
area, ice thickness, melting surface fraction, and downwelling SW and LW, all of which in HadGEM2-ES affect
the surface flux instantaneously, and can therefore be said to be quasi-independent.
Ice thickness does not appear in the surface flux formula directly; instead, the surface flux is expressed as a
function of the individual category thicknesses $h_{ice-cat}$. To estimate the ISF bias due to ice thickness, it is hence
necessary to sum over categories the ISF biases due to bias in each $h_{ice-cat}$. The estimation of model biases in
$h_{ice-cat}$, therefore, requires some discussion.
Given an estimated model bias in mean thickness $\overline{h}_{ice}^{'}$, it can be argued that the least arbitrary approach is to
estimate the model bias in each thickness category to be $\overline{h}_{ice}^{'}$ also (i.e. the thickness distribution is uniformly
shifted to higher, or lower values). However, this leads to unphysical results at the low end of the distribution; in
the case of a negative bias, it implicitly assumes the creation of sea ice of negative thickness; in the case of a
positive bias, it assumes that no sea ice of thicknesses between $0\ m$ and $\overline{h}_{ice}^{'}\ m$ exists.
Hence we use a slightly modified approach (Figure S1, supplementary material). The model bias in the lowest
thickness category is estimated to be $\overline{h}_{ice}^{'}/2$, equivalent to translating the top end of the category by $\overline{h}_{ice}^{'}$ but
allowing the lower end to remain at $0m$. The model biases in the other four categories are then estimated to be
$\overline{h}_{ice}^{'}\dfrac{a_{ice} - a_{1}/2}{a_{ice} - a_{1}}$, i.e. the translation is increased to ensure that the mean ice thickness bias remains correct.
Following this, we iterate through the categories, identifying grid cells where the bias is such that a negative
category sea ice thickness in the reference dataset is implied; in these cells, the bias is reduced such that the
reference thickness in that category becomes $0m$, and the bias in the remaining categories is increased
proportionally to ensure the mean sea ice thickness bias remains correct.
**Appendix B: Analysis of potential errors in ISF bias calculation**
The two principal sources of error in the ISF bias calculation method are examined in turn. Firstly, error in
correctly characterising the dependence of surface flux on a climate variable is estimated; secondly, error in
approximating the surface flux bias induced by this as the product of the surface flux dependence with the
model bias in that variable is estimated.
**B1 Error in calculating surface flux dependence**
To understand error in calculating dependence of surface flux on model variables, fields of the approximated
surface flux $g_{x,t}$ are compared to those of the real modelled surface flux $F_{sfc}$. The $g_{x,t}$ are found to capture
well the large-scale seasonal and spatial variation in surface flux, but are prone to systematic errors which vary
seasonally, indicated in Figure B1; firstly, a tendency to underestimate modelled negative surface flux in
magnitude from October-April by 13% on average; secondly, during May, an underestimation varying from 5-
20 Wm$^{-2}$; thirdly, a tendency to overestimate modelled positive surface flux from June-August by up to 10 Wm$^{-}$
$^{2}$.
Examining first the winter underestimation (demonstrated in Figure B1 a-c), it is found that for each model
month the relationship between estimated and actual surface flux is linear, with underestimation factors ranging
from 6 ± 1% in December to 17 ± 2% in April. This suggests that the cause lies in systematic underestimation of
the scale factor $\sum_{cat} \gamma_{ice-REF}^{cat} \left(1 - BR_{ice}^{cat}\right)^{-1}$. A possible cause is covariance in time between $\gamma_{ice-REF}^{cat}$ and $R_{ice}^{cat}$
within each month, particularly in the first ice category; during the freezing season, occurrence of high fractions
of ice in category 1, the thinnest category, would be expected to be associated with formation of new ice, and
correspondingly lower mean thicknesses of ice in this category, lower values of $R_{ice}^{cat}$ and higher values of
$\left(1 - BR_{ice}^{cat}\right)^{-1}$. A calculation using daily values of $\gamma_{ice-REF}^{cat}$ ranging from $0.1 - 0.5$, and daily values of $h_I^{cat}$
ranging from $0.2 - 0.5$m, predicts that this effect would in this case lead to an underestimation of 9% in the
magnitude of the surface flux, sufficient to explain all of the underestimation in October, December and
January, and most in November, February and March. This effect would produce a corresponding
underestimation of the rate of dependence of surface flux on downwelling LW radiation and ice thickness
throughout the freezing season. It is calculated that the downwelling LW component of the ISF bias is
underestimated by 0.6 Wm$^{-2}$ for the freezing season on average due to this effect.
Secondly, we examine the reasons for the underestimation of surface flux in May (Figure B1d-f), a pattern
unique to this month which is seen to be small in the central Arctic but to approach 20 Wm$^{-2}$ at the Arctic Ocean

coasts. A likely cause of this inaccuracy is the classification of grid cells as 'freezing' or 'melting' for entire months. During May, as has been seen, most model grid cells in fact cross from one category to the other; however, virtually all Arctic Ocean grid cells are classified as freezing for the month as a whole. The difference field between estimated 'freezing surface flux' and 'melting surface flux' is similar in magnitude and in spatial pattern to the underestimation field, being near-zero in the central Arctic but rising to 25 Wm$^{-2}$ close to the Arctic Ocean coasts. It is concluded that the actual model mean surface flux is much higher than that estimated near the coast due to these grid cells experiencing melting conditions from relatively early in the month. Although this error is not directly relevant to the results of this paper, as no unequivocal ISF biases were identified for May, it would have the potential to lead to overestimation of the dependence of surface flux on ice thickness, and underestimation of dependence on all other variables, as the upwelling LW flux is unable to counteract changes in surface forcing once the surface has hit the melting point.

Thirdly, we examine the tendency to overestimate surface flux during the summer (Figure B1g-i), an effect that displays a spatially uniform bias rather than a spatially uniform ratio, ranging from 5-15 Wm$^{-2}$ in July and August; the bias is smaller, and in the central Arctic negative, during June. A possible contributing factor to this bias is within-month covariance between ice area and downwelling SW; during July and August, both downwelling SW and surface albedo fall sharply, an effect that would tend cause the monthly mean surface flux to be overestimated. To estimate this effect, monthly trends in these variables were estimated by computing half the difference between modelled fields for the following and previous month. For July, an overestimation in surface flux of magnitude 5-15 Wm$^{-2}$ was indeed predicted in the Siberian seas, as well as the southern Beaufort and Chukchi Seas; however, in the central Arctic no overestimation was predicted, due to near-zero trends in ice area in the summer months. It is possible that some covariance between ice area and downwelling SW is nevertheless present in these regions, due to enhanced evaporation and cloud cover in regions of reduced ice fraction.

However, this effect would have no direct impact on the ISF biases because these are computed from monthly means of the model bias in one variable by the model mean in the other; hence, it is covariance between bias and mean that would induce inaccuracy in this case. By similarly approximating the trend in monthly mean model bias as half the difference between model bias in the adjacent months, the error in downwelling SW and ice area contributions were evaluated. Error in the downwelling SW term was found to be significant early in the summer, with an error of -2.7 Wm$^{-2}$ in June; error in the ice area term was found to be significant later in the summer, with errors of -1.7 Wm$^{-2}$ and -1.6 Wm$^{-2}$ in July and August respectively. However, the August error is small relative to the total ISF bias identified.

**B2 Error in characterising induced surface flux bias**

The surface flux dependencies, for each variable, are evaluated at a model state which is itself biased. This introduces an error in characterising the induced surface flux bias. For example, a component of the surface flux, net SW, is equal to $F_{SW\downarrow}\left(1-\alpha_{sfc}\right)$, and induced surface flux biases due to model biases in $F_{SW\downarrow}$ and

$\alpha_{sfc}^{'}$ would be calculated as $F_{SW\downarrow}^{'}\left(1-\alpha_{sfc}^{\mathrm{mod}}\right)$ and $F_{SW\downarrow}^{\mathrm{mod}}\alpha_{sfc}^{'}$ respectively. However, the sum of the two induced
surface flux biases will not be exactly equal to the true surface flux bias, $F_{SW\downarrow}^{\mathrm{mod}}\left(1-\alpha_{sfc}^{\mathrm{mod}}\right)-F_{SW\downarrow}^{obs}\left(1-\alpha_{sfc}^{obs}\right)$,
but will differ from it by $F_{SW\downarrow}^{'}\alpha_{sfc}^{'}$.
This apparent problem can be resolved partly by viewing the ISF method as a way not simply of estimating
model biases due to a particular variable, but of characterising them, i.e. by accepting that the quantity that we
are trying to estimate is itself somewhat subjective. Instead of requiring the ISF method to be correct, it is
required that it gives useful, physically realistic results. In the case given above, a sufficient condition is that
$F_{SW\downarrow}^{'}\alpha_{sfc}^{'}$ is small relative to $F_{SW\downarrow}^{'}\left(1-\alpha_{sfc}^{\mathrm{mod}}\right)$ and $F_{SW\downarrow}^{\mathrm{mod}}\alpha_{sfc}^{'}$, i.e. that the model bias in both downwelling
SW and in surface albedo is small relative to the absolute magnitudes of these variables.
More generally, the difference between the surface flux bias $F_{sfc}^{'}$ and the sum of the induced surface flux biases
$\sum_{i} v_{i}^{'}\,\partial g_{x,t}/v_{i}$ can be approximated by $\sum_{\substack{i,j \\ i\neq j}} v_{i}^{'}v_{j}^{'}\,\partial^{2} g_{x,t}/\partial v_{i}\partial v_{j}$ , a term that can be calculated relatively
easily as many of the derivatives go to zero. Averaged over the Arctic Ocean this term was small (below 1 Wm$^{-2}$
in magnitude) in most months of the year, but of significant size in October (3.6 Wm$^{-2}$), due to co-location of
substantial negative biases in downwelling LW and category 1 ice thickness in this month, indicating that the
true surface flux bias in this month may be substantially smaller (in absolute terms) than the -11.5 Wm$^{-2}$
obtained from summing the ISF biases.
Finally, the induced surface flux calculation implicitly assumes a linear dependence of surface flux on each
climate variable. However, this is not the case for the ice thickness, where higher-order derivatives do not go to
zero, and in some regions of thinner ice actually diverge. It is possible to quantify the error introduced by the
assumption of linearity by comparing the partial derivative $\left(A+BT_{b}\right)a_{cat}\left(1-BR_{cat}^{ice}\right)^{-2}\left(\sum_{cat} a_{cat}\right)^{-1}$ to the
quantity $\left(A+BT_{b}\right)a_{cat}\left(1-BR_{cat}^{ice}\right)^{-1}\left(1-BR_{cat}^{ice-REF}\right)^{-1}\left(\sum_{cat} a_{cat}\right)^{-1}$, where $R_{cat}^{ice-REF}=h_{I}^{OBS}/k_{I}+h_{S}/k_{S}$,
$h_{I}^{OBS}$ being climatological ice thickness in the reference dataset, in this case PIOMAS, and all other terms
defined as in Section 4. It can be shown that multiplying this quantity by the model bias produces the exact bias
in estimated surface flux that is being approximated by $\partial g_{x,t}/\partial h_{I}\left(h_{I}^{MODEL}-h_{I}^{OBS}\right)$. Hence the bias in the ice
thickness component induced by the nonlinearity can be calculated directly. It is found that the nonlinearity
causes the ice thickness component to be overestimated in magnitude by 0.7 Wm$^{-2}$ on average from October-
April, with a maximum overestimation of 1.9 Wm$^{-2}$ in November.
**Code availability**

The code used to create the fields of induced surface flux bias is written in Python and is provided as a supplement (directory 'ISF'). The code used to create Figures 1-9, as well as Figure B1, is also provided (directory 'Figures'). In addition, the routines used to estimate errors in the ISF analysis are provided (directory 'Analysis'). Finally, the code used to create Table 1 is provided (directory 'Tables'). A set of auxiliary routines used by most of the above are also provided (directory 'Library'). Most routines make use of the open source Iris library, and several make use of the open source Cartopy library.

**Data availability**

Monthly mean ice thickness, ice fraction, snow thickness and surface radiation, as well as daily surface temperature and surface radiation, for the historical simulations of HadGEM2-ES, is available from the CMIP5 archive at https://cmip.llnl.gov/cmip5/data_portal.html.

NSIDC ice concentration and melt onset data can be downloaded at http://nsidc.org/data/NSIDC-0051 and http://nsidc.org/data/NSIDC-0105 respectively.

PIOMAS ice thickness data can be downloaded at http://psc.apl.uw.edu/research/projects/arctic-sea-ice-volume-anomaly/data/.

ERAI surface radiation data can be downloaded at http://apps.ecmwf.int/datasets/data/interim-full-daily/levtype=sfc/.

ISCCP-FD surface radiation data is available at https://isccp.giss.nasa.gov/projects/browse_fc.html.

CERES surface radiation data is available at https://climatedataguide.ucar.edu/climate-data/ceres-ebaf-clouds-and-earths-radiant-energy-systems-ceres-energy-balanced-and-filled.

**Acknowledgements**

 This work was supported by the Joint UK BEIS/Defra Met Office Hadley Centre Climate Programme (GA01 101), and the European Union's Horizon 2020 Research & Innovation programme through grant agreement No. 727862 APPLICATE. MC was supported by NE/N018486/1.

Thanks are due to Richard Wood and Ann Keen for helpful comments on multiple versions of this study.

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

| Month | Downwelling SW | Downwelling LW | Ice thickness | Ice area | Melt onset occurrence | Total induced surface flux bias | Radiative flux bias | ISF residual | CERES-ERAI net radiation difference |
|---|---|---|---|---|---|---|---|---|---|
| Jan | 0.0 | -6.5 | -4.3 | 2.7 | 0.0 | -8.1 | -1.6 | 3.3 | -12.2 |
| Feb | 0.0 | -4.8 | -2.8 | 2.3 | 0.0 | -5.3 | -10.4 | 4.6 | -12.5 |
| Mar | 0.1 | -3.8 | -2.0 | 2.0 | 0.0 | -3.7 | -10.5 | 5.9 | -12.0 |
| Apr | 0.4 | -4.4 | -1.4 | 1.4 | 0.2 | -4.2 | -12.2 | 7.6 | -9.7 |
| May | 2.1 | -4.8 | -0.6 | 0.0 | 0.1 | -3.2 | -3.7 | 0.5 | -3.5 |
| Jun | -8.3 | 7.4 | 0.0 | 1.7 | 11.4 | 12.2 | 27.8 | -15.4 | -4.2 |
| Jul | -13.6 | 8.0 | 0.0 | 3.7 | 3.3 | 1.4 | 5.1 | -3.9 | -16.9 |
| Aug | 0.5 | -3.3 | -0.1 | 9.6 | 1.9 | 8.5 | 8.6 | -0.0 | -12.9 |
| Sep | 3.3 | -7.1 | -0.7 | -0.7 | 0.0 | -5.2 | -0.2 | -4.8 | -2.4 |
| Oct | 0.9 | -5.7 | -4.2 | -1.8 | 0.0 | -10.8 | -14.4 | 3.4 | -4.6 |
| Nov | 0.0 | -5.4 | -8.3 | -0.3 | 0.0 | -14.0 | -21.6 | 8.1 | -11.7 |
| Dec | 0.0 | -6.4 | -6.4 | 2.4 | 0.0 | -10.4 | -14.9 | 4.1 | -12.2 |

**Table 1. Surface flux biases induced by model bias in 5 different variables in HadGEM2-ES (Wm$^{-2}$), with**
**CERES used as reference dataset for the radiative components. Total ISF bias and total net radiative flux**
**bias relative to CERES are shown for comparison, as well as their residual; the difference between net**
**radiative flux bias as evaluated by CERES and ERAI is also shown. A positive number denotes a**
**downwards flux, and vice versa.**

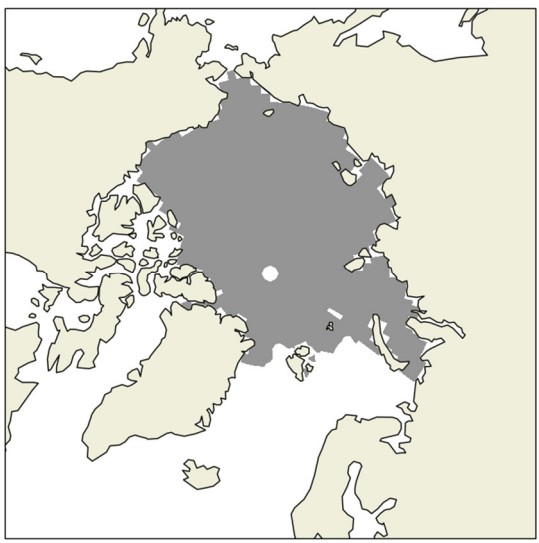

2     **Figure 1. The Arctic Ocean region used in the analysis, defined as the area enclosed by the Fram Strait, Bering Strait**
3     **and the northern boundary of the Barents Sea.**

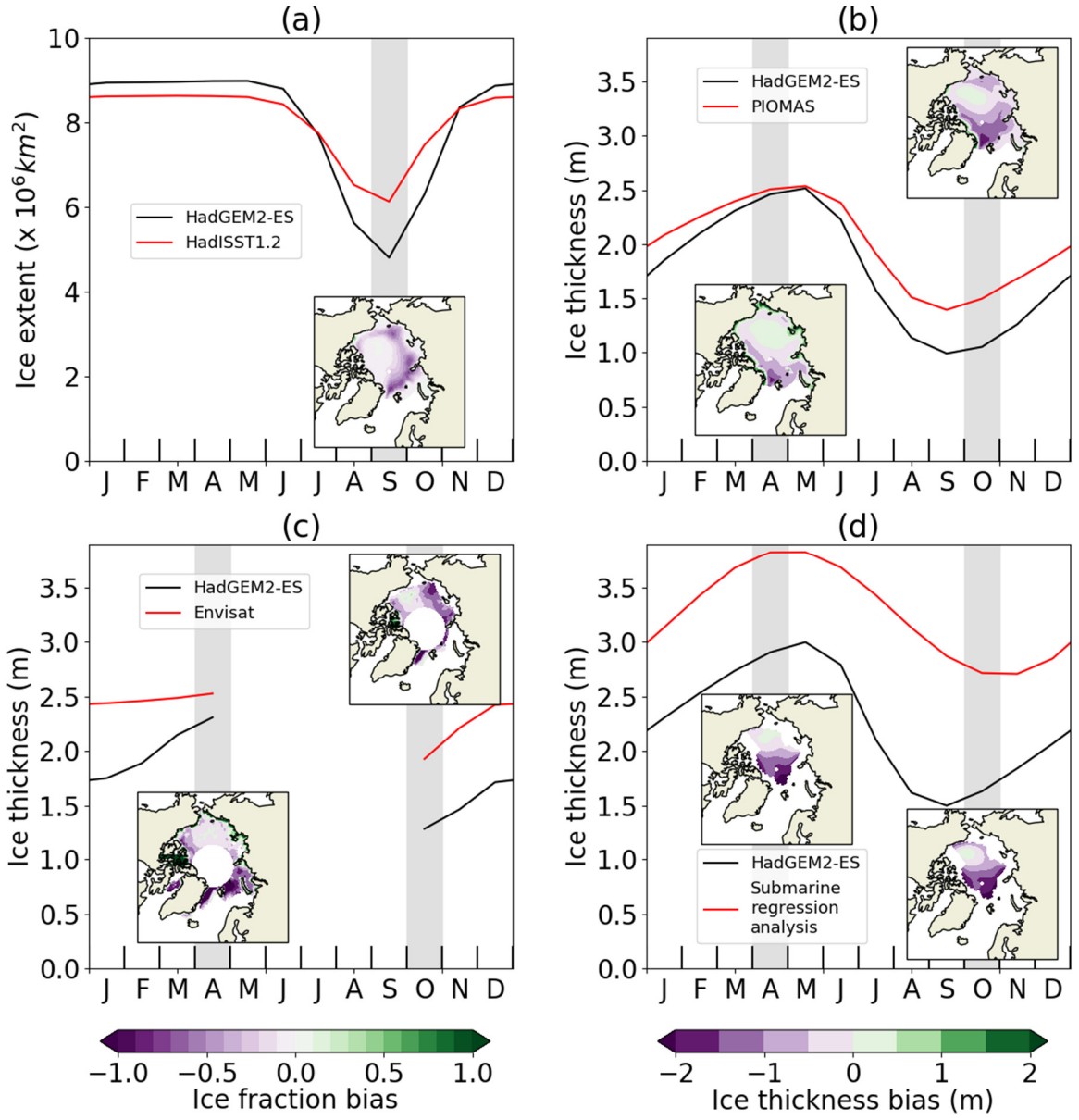

**Figure 2. (a) HadGEM2-ES 1980-1999 mean Arctic Ocean ice extent, compared to HadISST1.2 1980-1999, with**

**September ice fraction bias map; (b-d) HadGEM2-ES ice thickness compared to (b) PIOMAS 1980-1999, (c) Envisat**

**1993-1999 and (d) submarine datasets from 1980-1999 over respective regions and periods of coverage, with April**

**and October ice thickness bias maps. For each seasonal cycle plot, the model is in black and reference datasets in red.**

**In (c), data is not plotted from May-September due to the region of coverage being very small.**

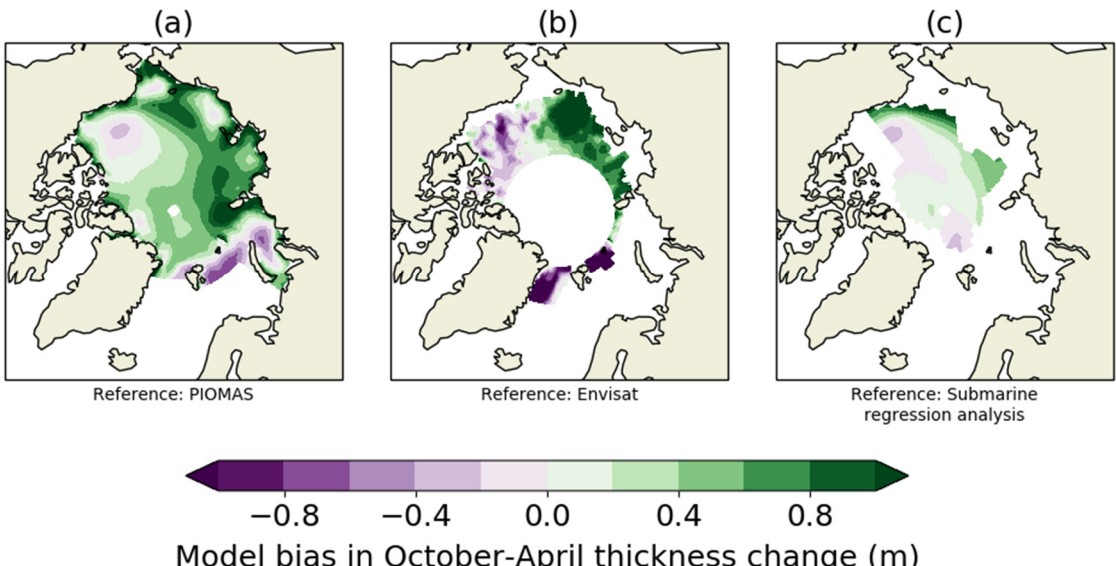

**Figure 3. HadGEM2-ES model bias in ice thickness change from October-April compared to (a) PIOMAS 1980-1999;**

**(b) Envisat 1993-2000; (c) submarine regression analysis 1980-1999. In each plot, the model period used matches the**

**period of the reference dataset. Differences are taken as model-reference so that areas of green (purple) correspond**

**to areas where the HadGEM2-ES model simulates too much (not enough) sea ice growth through the winter.**

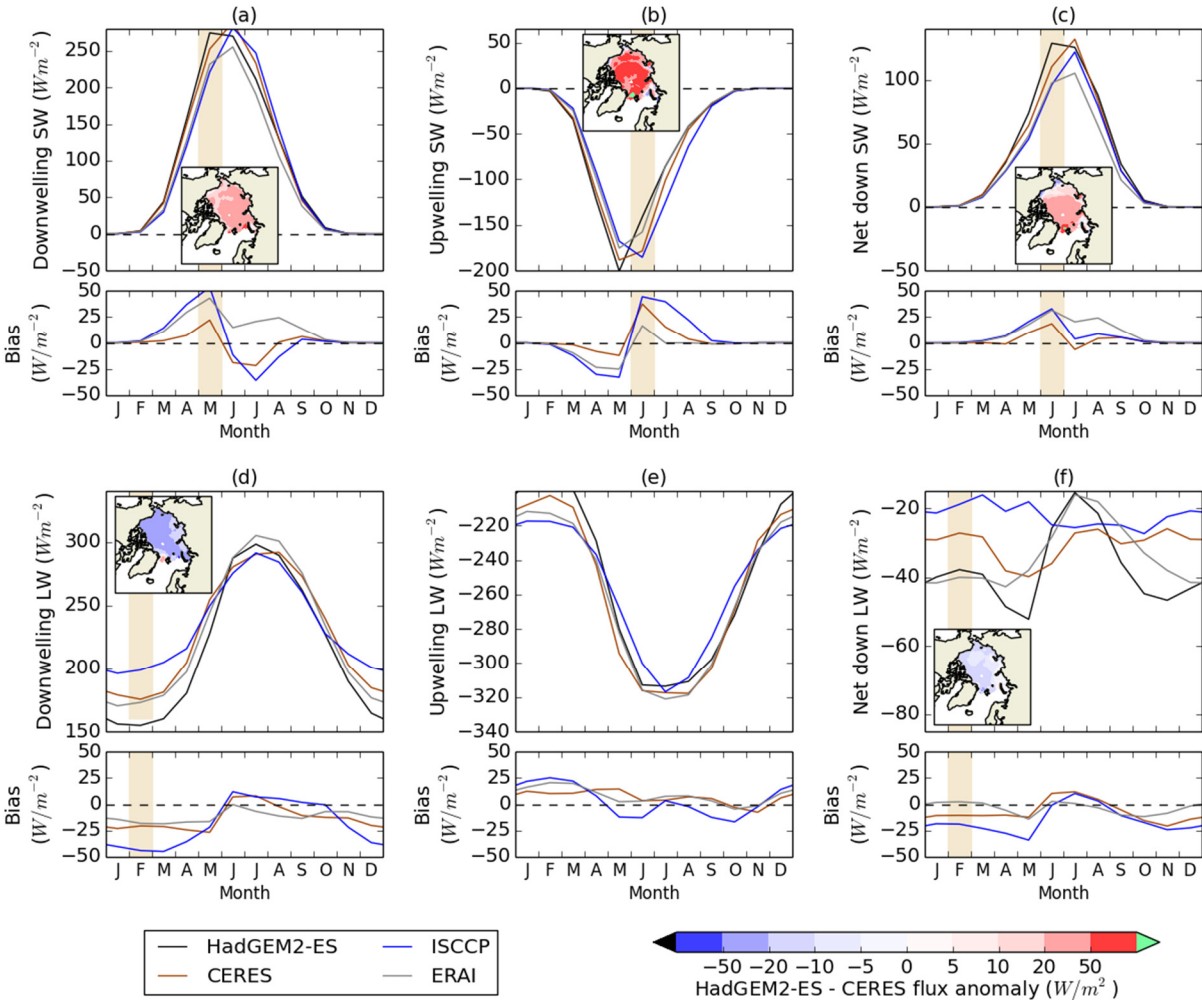

**Figure 4. (a) Downwelling SW, (b) upwelling SW, (c) net down SW, (d) downwelling LW, (e) net**
**down LW, for HadGEM2-ES 1980-1999 over the Arctic Ocean region, compared to CERES 2000-2013, ISCCP-D**
**1983-1999 and ERAI 1980-1999. Upper panels show absolute values; lower panels show model bias relative to each**
**respective dataset. For all fluxes, a positive number denotes a downward flux and vice versa. Maps of flux bias**
**relative to CERES are shown for downwelling SW in May, upwelling and net down SW in June, and downwelling**
**and net down LW in February.**

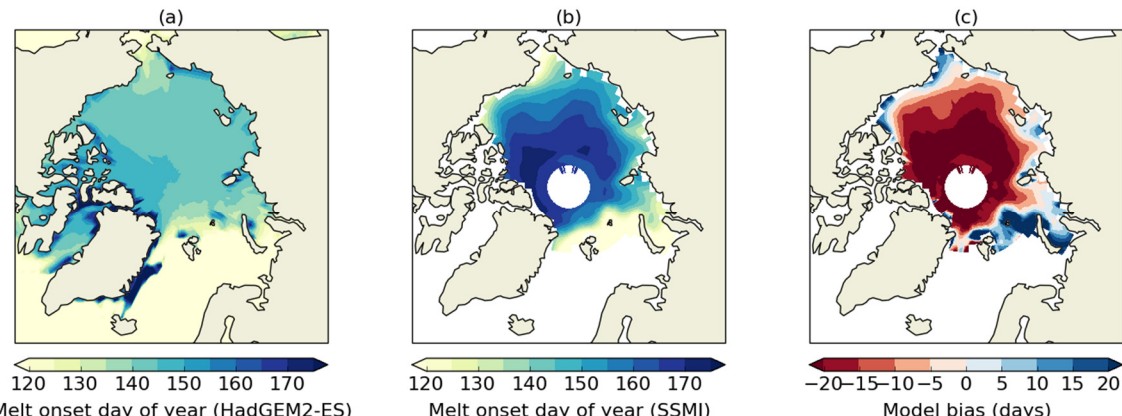

**Figure 5. Average date of year of surface melt onset, 1980-1999, (a) as modelled by HadGEM2-ES, (b) as measured by SSMI observations. (c) shows model bias.**

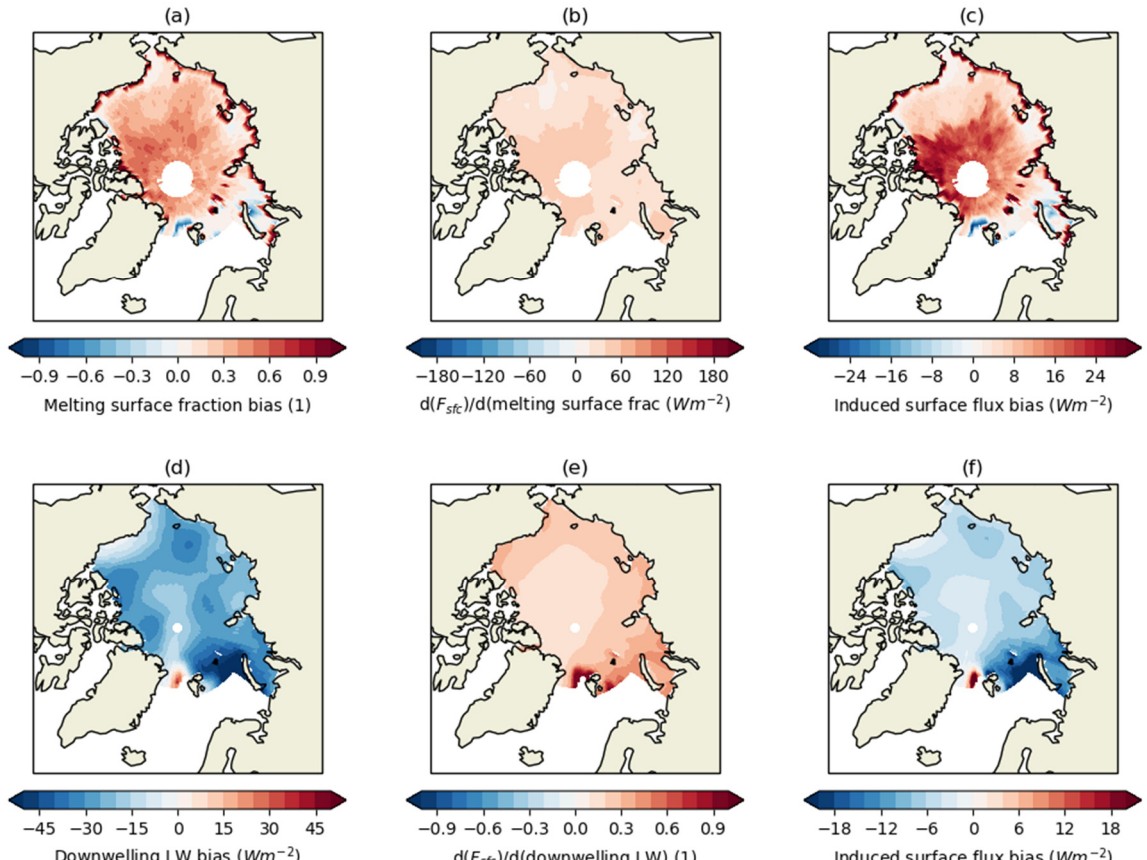

Figure 6. Demonstrating the calculation of fields of surface flux bias due to model bias in melting surface
fraction (a-c) and downwelling LW (d-f). The left-hand column shows model bias in each variable; the
middle column the local rate of dependence of surface flux on each variable as calculated above; the right
column the induced surface flux bias, calculated as the product of these two fields. The first historical
simulation of HadGEM2-ES is used for the illustration.

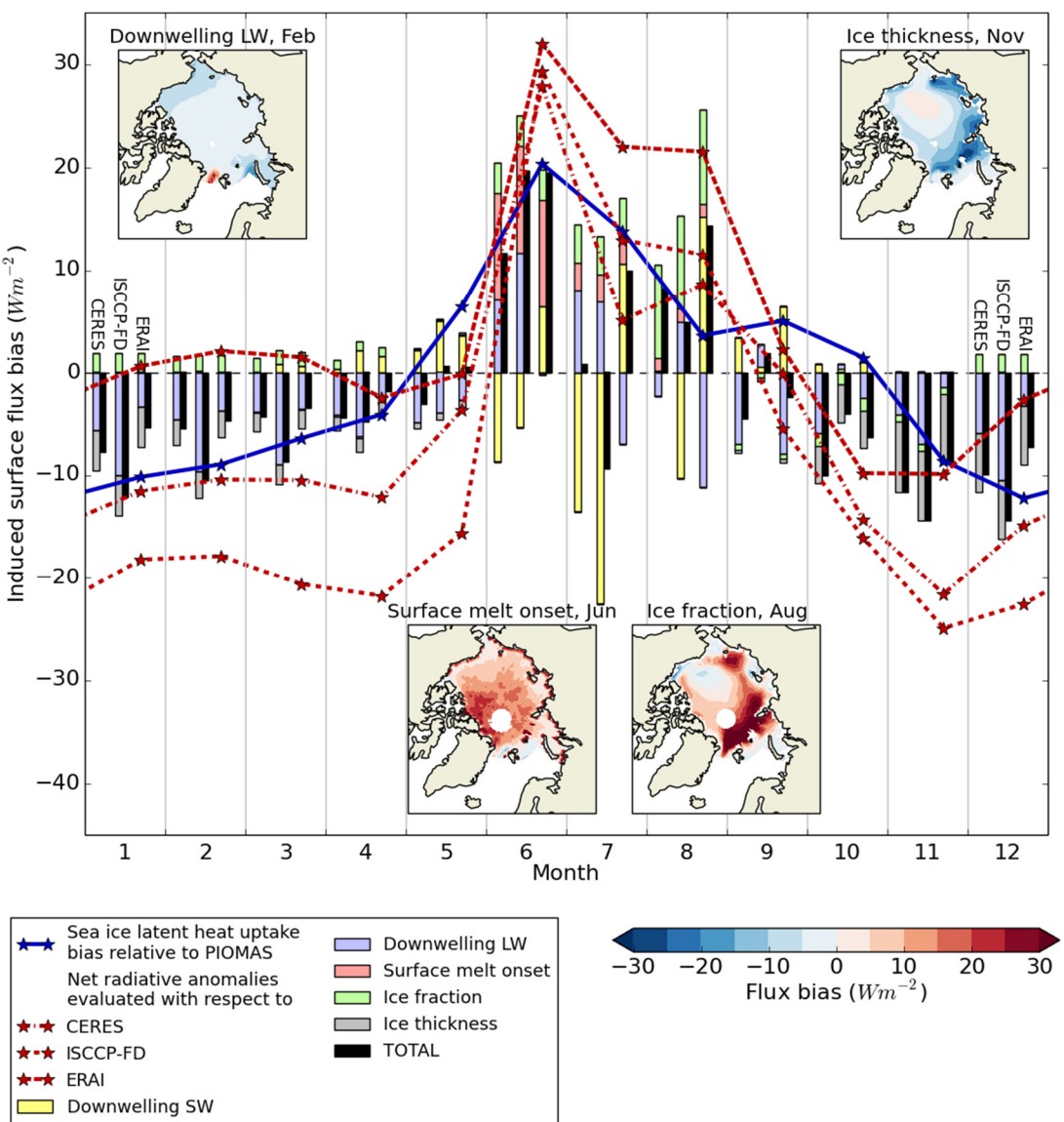

**Figure 7. Surface flux bias induced by model biases in ice fraction, melt onset occurrence, ice thickness, downwelling SW and downwelling LW respectively, for the Arctic Ocean region in HadGEM2-ES, 1980-1999. Total ISF bias is indicated in black bars. For each month, induced surface flux biases are estimated using in turn CERES, ISCCP-FD and ERAI as radiation reference datasets, from left-right. Sea ice latent heat flux uptake bias relative to PIOMAS is indicated in black. Net radiative flux biases relative to CERES, ISCCP-FD and ERAI are indicated in brown. Spatial patterns of induced surface flux bias for four processes in key months, with CERES as reference dataset, are displayed beneath and above.**

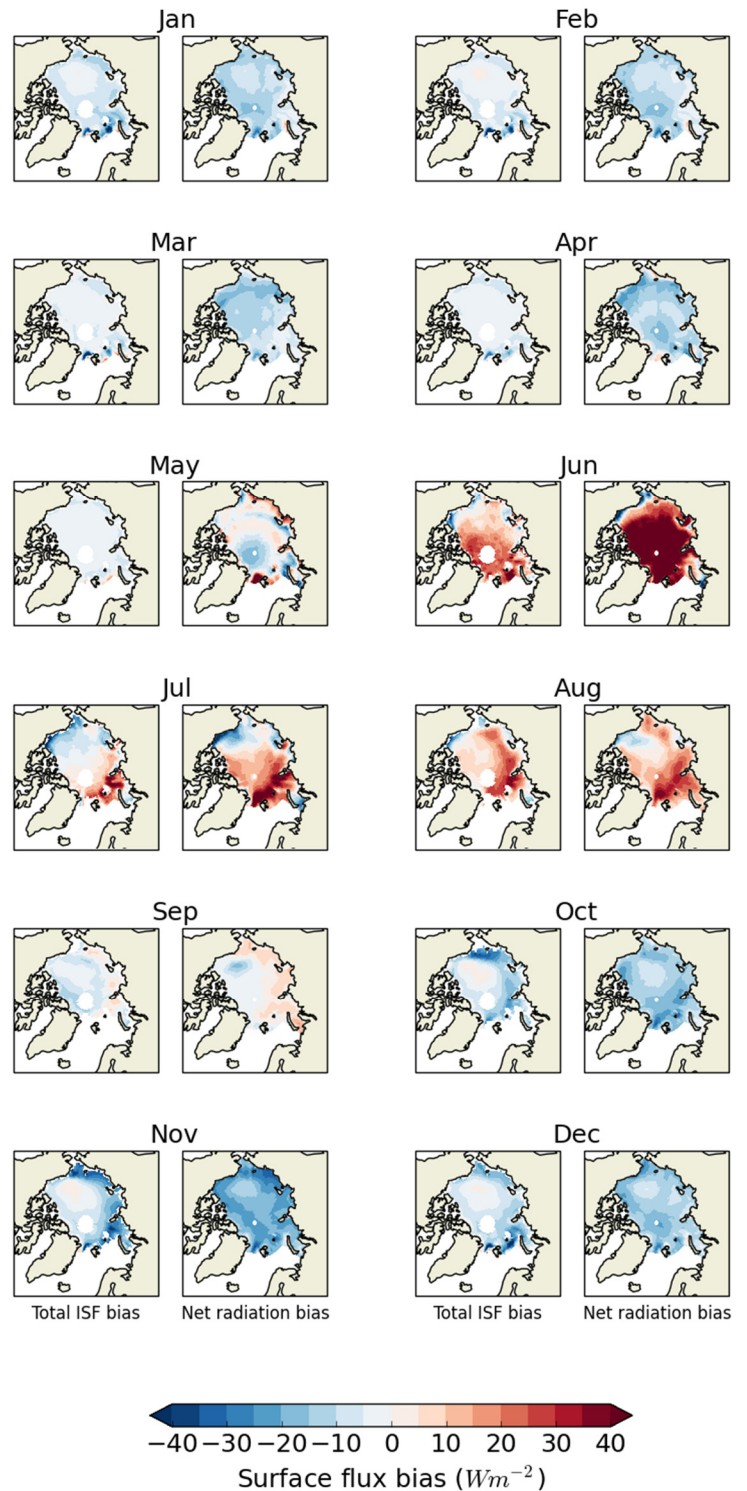

**Figure 8. Comparing fields of total ISF bias (left) to net radiation bias (right) relative to CERES for each month of**

**the year, for the four historical members of HadGEM2-ES, 1980-1999.**

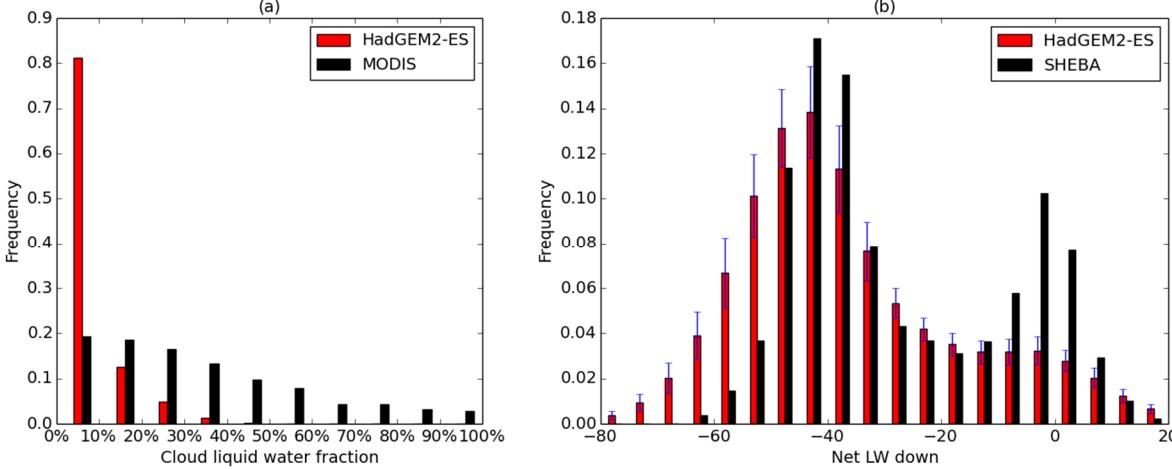

Figure 9. Frequency distributions of (a) October-April cloud liquid water percentage in HadGEM2-ES compared to MODIS observations, for the Arctic Ocean region; (b) December-February surface net downwelling LW in HadGEM2-ES in the SHEBA region, compared to the values observed at SHEBA.

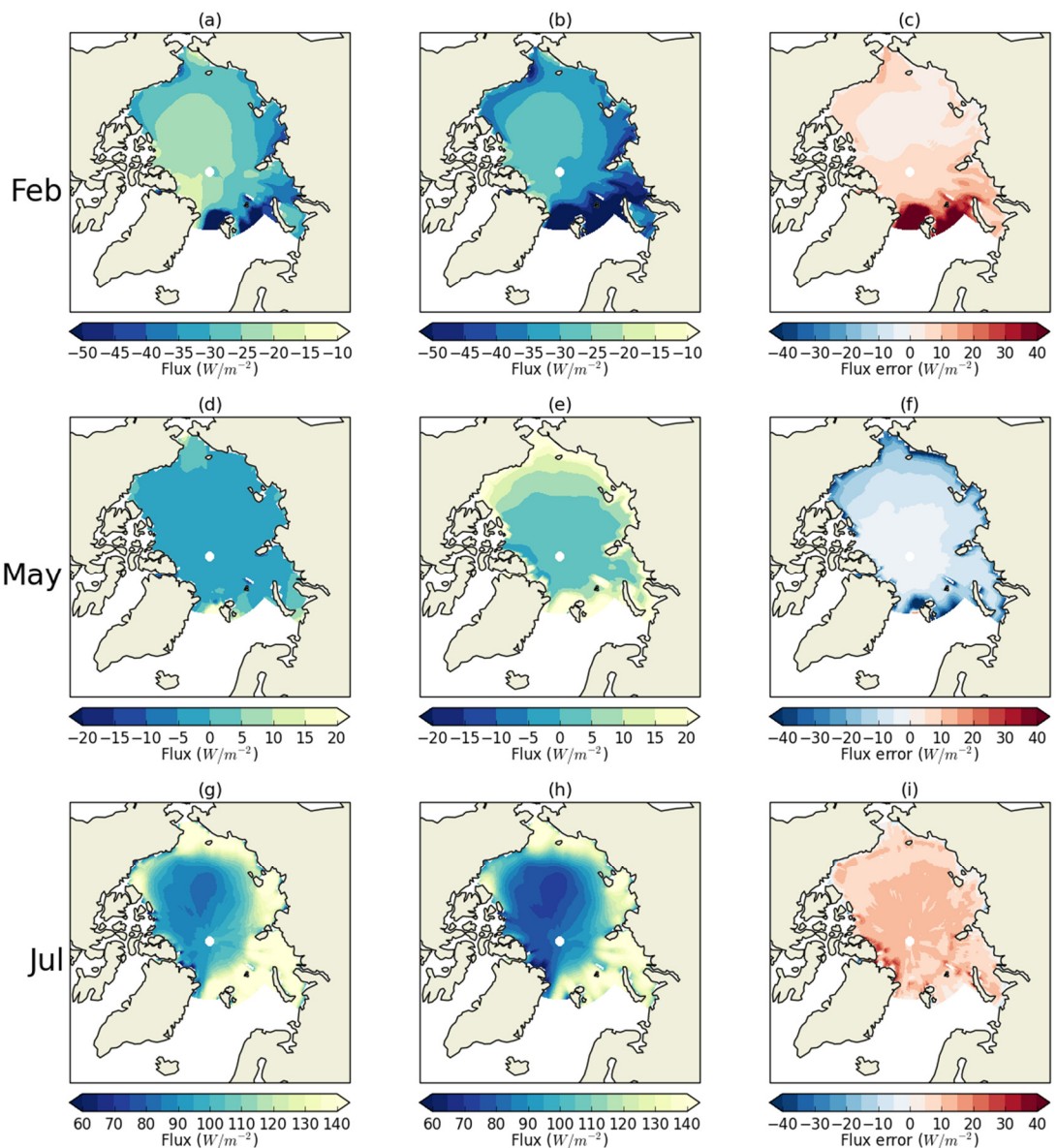

**Figure B1. Illustrating approximated (left) and actual (centre) model net surface flux, as well as the approximation error (right), in (a-c) February; (d-f) May; (g-i) July, for the period 1980-1999 in the first historical run of HadGEM2-ES.**