# Peer review of "Induced surface fluxes: A new framework for attributing"

_The Cryosphere, 2018_

## Referee Comment (RC1) · Anonymous Referee #1 · 16 May 2018

West et al propose a new analysis framework to understand model biases in Arctic sea ice which they apply to HadGEM2-ES, a model with known biases in sea ice characteristics.

The attribution of climate model errors in the sea ice zone is a very important open topic and the paper provides original and likely efficient means to evaluate such errors.

The main problem I think is writing, which I found often imprecise, and renders a proper evaluation of the paper difficult.

In particular, the methods absolutely require clarification and should use better and simpler terminology. Because I did not fully get the methods, it was thereafter really

complicated to follow, in particular the discussion and conclusions.

A second requirement to make this paper acceptable is to early on in the result section to explain that the induced surface flux method works - eg. to describe how well the different methods to compute surface flux biases converge. Now this is done here and there, and I have constantly been doubting of the quality of the methods, because of the absence of such evaluation.

A third thing I would have enjoyed to see is a specific discussion of how the ice-albedo and growth-thickness feedbacks can be diagnosed from the method. It is claimed in the abstract that your method can separate these effects, and I am in trouble to see how that statement is presently supported in the text. I can guess feedbacks are acting from Fig. 6, but I think this topic deserves a bit more to support the claim made in the abstract.

I have also not understood why energetic errors of oceanic origin have been ignored from the discussion, especially in the North Atlantic sector of the Arctic - where there is a low bias.

Finally, the authors claim in the conclusions that they can "quantify" the origin of errors, but apart from Fig. 6 (which I liked a lot), I did not really see a quantification of the errors. Is that quantification the main point - or is it the consistent comparison of the different sources of error? Also, it was difficult to ultimately figure out whether biases in external forcings or in the sea ice model are the ultimate cause of the biases. Is your method capable to tell after all ?

A last general comment - the logics of the arguments should be better presented.

I am pretty confident that - if these presentation issues are seriously addressed by the team of coauthors, this will make an excellent contribution to their favourite cryospheric journal.

TCD
A few specific comments.

**\* I have tried to understand what the generic approach is. Here is what I have understood. The present presentation is too lengthy, misses the essential elements and overdiscusses details. A synthetic view is missing. There are three means to evaluate errors in surface energy budget (I have understood two of them)**

1) The direct computation of surface flux bias, i.e. the difference between simulated and observed surface flux (or one of its components)

2) The induced surface flux bias, which is the contribution of bias in a specific variable to surface flux bias, namely calculated as  $\Delta Fx = dFx / dx \Delta x$  (mod-obs).

To evaluate derivatives, the SEB is simplified using two different approximations during the cold and warm seasons, based on ideas from Thorndike et al 1992.

I don't think there is a need to calculate those derivatives in the body of the paper.

If the derivatives are well calculated and if the non-linearities are not too important, the sum of  $\Delta$ Fx should hopefully approach the surface flux bias.

3) The third diagnostic is "the sea ice latent heat flux uptake anomaly implied by the ice volume anomalies relative to PIOMAS".

I have tried to figure out what the authors mean, but I did not really managed. The wording is not precise enough for the reader to what is meant by this and what is gained by comparing that to the surface flux biases. I guess "latent heat flux" is confusing in the context of the surface energy budget. But whether that thing is a heat storage anomaly divided by time or something else, I don't know. Maybe an "ice thickness bias converted to Joules" or "an energetic equivalent ice thickness bias" ?

Besides an explanation of what it means, we would need an explanation of what should be taken from that diagnostic.

TCD
It is important to clarify this point because a lot of the argumentation was based on that.

\* The two methods to compute the surface flux derivatives is called "a model". I think it is a "computation method". It is actually inspired from Thorndike et al (1992) - which should be acknowledged - and maybe from earlier works in EBMs. What you are doing is to derive the surface energy budget wrt anything.

---

## Short Comment (SC1) · 8 Jun 2018

Review of West et al.: Attribution of sea ice model biases to specific model errors enabled by new induced surface flux framework

by F. Massonnet

Note: I have not read the other (public) referee comment in order to ensure a maximum of independence in my review.

In this paper, Alex West and colleagues introduce a framework to decompose the bias in sea ice surface energy budget simulated by the HadGEM2-ES GM and estimate

the relative contributions from individual terms (thickness bias, concentration bias, melt onset bias, and atmospheric forcing). To a first order, the net surface flux is proportional to the ice volume change, thus this method is used to understand possible model-data mismatch in terms of simulated sea ice cover. The HadGEM2-ES model is found to have a negative thickness bias in summer due to a bias in melt onset date in spring, and an overestimation of the seasonal cycle of thickness due to an underestimation of downwelling long-wave flux in winter leading to excessive ice growth.

I find this work interesting. To my knowledge, this type of approach (by modelling the model dependence of fluxes on different state variables) has not been introduced before. Its main interest is that no extra experiments are required, meaning that the same analysis can be applied on large model ensembles in a relatively straightforward way. I see a lot ot potential for this work, especially in view of the upcoming CMIP6.

That said, I have a few reservations that should be addressed by the authors, which I list below. I do think that addressing these points can eventually contribute to produce a very valuable manuscript.

1) It is not always easy to follow the authors methodology. I see the general idea behind the approach: expressing the net flux to the ice as a function of state variables, then linearizing around a reference state to obtain the flux bias resulting from the bias in one of the model components. However, I could surely not reproduce the results myself, just based on the text. I appreciate the efforts to publish the code in Supplementary Material, but the text itself should have all elements. For example, I do not understand how the bias in F_sfc attributed to error in melt onset is derived (i.e., from eq. A6 to 4). Furthermore, the attribution of flux error to melt onset error seems to not be a function of the melt onset itself, but rather a function of concentration difference. This is confusing: melt onset is defined by the time of the day where surface melting commences, and the right hand side of Eq. 4 does not display surface melting terms. At some point it came to my mind that the authors were perhaps using "melt onset" for "ice retreat", but I'm not sure. In all cases, this is confusing.

2) Besides the need for clarity in the methodology, a key question is to what extent the assumption of linearity holds, in particular for what bias range Eqs 3 and 4 would be valid. Will the methods work for models with very large biases? Another point is that this linearization involves the use of Eqs 1 and 2, that are themselves derived using linearity assumptions. I trust that the approach is valid, because the sum of individual contributions (Fig. 6) seems to match the flux errors from datasets and from volume estimates, but a quantification of this match should be done (perhaps by calculating residuals). Overall I find that the authors have not discussed the validity of this assumption, and this is critical given how non linearly the ice behaves.

3) I would also like to see if the method is robust to internal variability. Could the authors take one or several of the four other members of the HadGEM2-ES model and run the same analysis? In other words, is the 1980-1999 period long enough to identify and attribute the biases?

4) The authors have not cited an important study: Holland et al., 2008 (doi:10.1007/s00382-008-0493-4). In that study, the inter-model scatter in the sea ice mass budget (present-day conditions) is shown to be explained by the way models absorb shortwave radiation. This is directly relevant to the study here, and I think the authors should go through the Holland et al. study to position their results with respect to theirs. In particular the claim that turbulent fluxes are of relatively minor importance relative to radiative fluxes in setting the surface energy balance (Fig. 2, and p. 5 line 11-16) should be put in perspective with that study. As the Arctic sea ice mean state changes, turbulent fluxes appear to be of increasing importance.

5) The authors should prove, with a figure, that the model developed in the appendix is good enough to do the investigations. Could they plot, for one or several grid cells and one or several freezing seasons, the reconstructed flux F_sfc (Eq. A4) and the actual flux from the model? A quantification of the correspondence would be a plus.

6) Nothing is said about the treatment of snow in the HadGEM2-ES model. How many

snow layers are there, what is snow conductivity, etc.?

7) The authors repeatedly use the word "anomaly" to describe the difference between modeled and reference quantities, but I would avoid this word and use "bias" or "error" instead. To me, an "anomaly" is used to described the deviation of a signal with respect to its own mean

8) I'm unclear about whether ocean surface temperature biases are accounted for in the analysis. From Fig. 6, it looks like they are not. On the other hand, p. 5 lines 6-10 seem to suggest that the Arctic Ocean is critical in setting the ice energetic balance (and this is also seen in Keen et al (https://link.springer.com/article/10.1007/s00382-013-1679-y, their Fig. 4). So, I'm puzzled: is the contribution of oceanic surface temperature bias taken into account or not in the analysis?

Minor comments (page-line)

1-18 - "countered by a counteracting" is a bit odd. 1-24 - "from 1986-2015" –> from 1986 to 2015 1-28 - Along with an earlier melt onset date, you can mention that freeze up has been delayed: Stammerjohn et al., 2012, their Fig. 2 (doi:10.1029/2012GL050874) 1-29 - "whoseloss" –> "the loss of which" ?

2-1 - Evaluation against observations of volume is quite impossible (even extent observations are not direct observations), so I would use "observational or reanalysis reference datasets" 2-18 Instead of "anomalies" I would use "biases".

3-24 The period 1980-1999 is used for evaluation, because it "predates the rapid sea ice loss". Why is it a problem to have a period with strong trend in the analysis? Is it expecting that the SEB would change too rapidly during a period with strong trends? Would the analysis be robust if the model output was evaluated on a distinct and later period (2000-2015 for instance, using historical + RCP8.5 runs). Please elaborate. 3-35 Reanalysis data also suffer from biases because of errors in atmospheric forcing, this could be stated as well.
5-11/16 Can the authors explain exactly what they mean by "Heat flux due to snowfall". The presence of snow affects heat conduction fluxes and acts to reduce bottom growth, is that what the authors are talking about? 5-26 I assume $h\_I$ and $h\_s$ refer to in-situ / actual thickness (this is the one that matters for vertical thermodynamics). It would be good to mention that here, as there is usually a lot of confusion between that quantity and the grid cell average thickness. 5-24 Eq. 1: Maybe I missed it, but what is the value for ice albedo? Does albedo depend on the ice state? 5-27 The subscript for snow thickness is "S" here while it is "s" elsewhere 5-28 The symbol for albedo is \alpha_I while in the equation it's \alpha_ice 5-30 Eq. 2: the big "dot" is a bit disturbing, it makes me think at a scalar product. I would use a simple dot or no dot at all. 5-6/10 The sentence "Because of this, although advection-derived ocean heating..." is unclear to me. First, can you demonstrate the oceanic heat convergence (that is not accounted for in your framework) is a small contributer to volume changes? Second, I do not follow the logical articulation with the next sencen "Hence the surface energy...". Please clarify. In the same line, reading the recent paper by Lei et al. (2018, doi:10.1002/2017JC013548) could be useful to add up to the discussion.

6-1 Please give the albedo values used. 6-3 "summarises" –> "summarise" 6-10 Eq. 3: please describe the meaning of the terms of the equation. In particular, what is $h\_I\_eff$? It is necessary to have this information in the text somewhere. 6-23 The partial derivatives are to be evaluated at a reference state, and I understand here that a mid-point between observations and model is taken ("Where observational datasets were available, the reference quantities in the partial derivative fields were calculated as model-observation means"). The authors should explain why it was done this way. I assume that the reconstructed flux error would be mathematically closer to the actual error than if the reference was taken as either the model or the osbserved value. 6-30 The paragraph starts by saying that 4 ensemble members were run, but Fig. 3 only shows one. Can you clarify?

8-28 The word "save" should be removed, I think

10-25/28 Can you go a bit more quantitative here? From Fig. 6, the residual of the analysis can be calculated as the sum of individual contributions (the stars) and the actual flux error. 10-31/32 The surface albedo feedback is not just a sea ice concentration thing. The melting of snow, the thinning of the ice are also key players in the surface albedo reduction, even though ice concentration remains unchanged. Wouldn't it make more sense to include these factors as well in the definition of surface albedo feedback?

11-15 The sentence "Hence the melt onset anomaly, acting alone, would induce a seasonal cycle of sea ice thickness both lower, and more amplified, than that observed..." is unclear, especially regarding the "lower" part. Can you please rephrase?

12-1 "concludethat" –> conclude that

Fig. 3. I'm puzzled by panel (a). Sea ice extent seems small. Is that because the domain "Arctic Ocean" is restricted to the seas of Fig. 1? In other observational records, like NSIDC, winter sea ice extent is more in the 14-16 million km2 range.

---

## Author Response (AR1)

**Description of changes made to 'Attribution of sea ice model biases to specific model**
**errors enabled by new induced surface flux framework'**

In this document, changes made to the manuscript arising from the reviews are described.
The original point-by-point response to the reviews, and a version of the manuscript with
changes tracked, are appended.

**1.  Fundamental changes to the analysis**
In response to a suggestion by Reviewer 2, fields of surface flux estimated by the formulae
described in Section 2.3 of the original paper were compared to modelled actual surface flux
fields. While the estimated fields captured well the seasonal and spatial variation in surface
flux, large discrepancies in the absolute values were apparent (30-40 $Wm^{-2}$ in some
months). As a result of these, the methods of estimating surface flux were refined in the
following ways:

- Surface flux contributions were estimated for each ice thickness category separately,
and then summed (as opposed to using the mean ice thickness across all
categories);
- The reference temperature about which the Stefan-Boltzmann relation is linearised
was allowed to vary in space and time (it was previously uniformly 0°C). For each
grid cell, this reference temperature was set to the monthly mean surface
temperature.
- A representation of the turbulent fluxes was added to the formulae.

The biases in estimated surface flux were greatly reduced by this method. The remaining
biases, and their implication for the results, are discussed in the new Appendix A, which
examines potential errors in the induced surface flux analysis.

Although the results of the analysis were not qualitatively changed by the new methods, the
contribution of the ice thickness bias during the winter to the surface flux bias was somewhat
increased, probably because the greater efficiency of ice production for thin ice is captured
more effectively by using the full thickness category information.

Because the new surface flux formulae use much more detailed information about the model
diagnostics, the approach of calculating surface flux dependency on each variable has
changed. Whereas in the previous version, the model-observation mean was used to
calculate partial derivates (i.e. surface flux dependency on variable), in the new version only
the model state is used to calculate this. Implications of this are also discussed in the new
Appendix A.

**2.  Major additions and corrections by section**
The structure of the paper has been altered slightly. The description of the induced surface
flux (ISF) analysis has been moved later in the paper, to a new Section 4, after the model
sea ice and surface radiation evaluation in Section 3. The derivation of the formulae is now
described briefly within this section, rather than in an appendix. Hence the new structure is

Below, major additions and corrections are described section by section (minor corrections
are summarised in Section 3 of this document).

**2.1 Abstract and introduction**

In these sections, the presentation of the logic behind the ISF method has been altered, in
part inspired by the changes described in Section 1 above. It is hoped that as a result the
presentation is improved and clarified.

The argument is stated again here for ease of reading: at any model point in space and time,
we approximate model surface flux as a function of climate variables that affect the surface
flux on timescales shorter than that on which they affect each other. In this way, by taking
partial derivatives, we characterise at each point in space and time the rate at which surface
flux depends on any climate variable. Multiplication of the resulting field by an estimate of
model bias in that variable therefore produces a field of estimated surface flux bias induced
(on a near-instantaneous timescale) by the model bias in that variable. These fields can be
averaged in space and time to give large-scale estimates of surface flux bias, bypassing
nonlinearities present in the relationship between surface flux and climate variable. This
allows the proximate causes of surface flux bias (and hence bias in sea ice growth and melt)
to be directly quantified.

The argument above is presented fully in the new Section 4, but is summarised in the
Introduction, and more briefly summarised in the abstract. It is stated more carefully that the
ISF method can diagnose only proximate causes of surface flux bias, as pointed out by
Reviewer 1.

**2.2 Model and observational data**

In the model description, the sub-gridscale thickness distribution of HadGEM2-ES is now
described more carefully, as it is now key to the analysis. The snow thermodynamics is also
described in more detail, as requested by Reviewer 2. The motivation for the use of the
period 1980-1999 is also described more fully, also as requested by Reviewer 2.

No major corrections are present in the observational data description section.

**2.3 Sea ice state and surface radiation evaluation**

For this section, and all subsequent sections, the Arctic Ocean region has been refined to exclude more of the Barents Sea, where processes are considered sufficiently different to the rest of the Arctic Ocean as to render the assumptions used in the ISF analysis questionable (for example there is high oceanic heat convergence here, and ice concentrations are low or zero year-round). Figure 1, which maps this region, has been changed accordingly.

In addition, unless stated otherwise, all model results are now given for the ensemble mean of the 4 historical simulations of HadGEM2-ES (rather than for the first historical simulation only, as had been the case for the previous version).

As a result of these changes, many of the sea ice thickness and surface radiation biases quoted in Section 3 are different. The previous Figures 3-5 (presenting the evaluations) have also been overhauled accordingly, and are now numbered Figures 2-4 (as discussed below, it was decided that the previous Figure 2 was superfluous and has been removed).

The qualitative conclusions of the evaluation are unchanged: sea ice thickness is too low in the annual mean and too amplified: net SW is too high and upwelling SW too low in the summer, net LW and downwelling LW are too low in the winter.

**2.4 ISF method description**

This section has been completely rewritten, reflecting the changes in the methodology described above. Firstly, the use of the word 'bias' is defined (as requested by Reviewer 2, all instances of 'anomaly' the paper have been replaced with 'bias'). This is followed by a discussion of the relationship between sea ice mass balance, surface flux and oceanic heat convergence, clarified along the lines suggested by both reviewers, and with additional evidence cited that in both HadGEM2-ES and in the real world, oceanic heat convergence is a minor contributor to sea ice mass balance in the Arctic Ocean interior.

With the link between surface flux and sea ice mass balance established, the approach of the study (see 2.1 above) is set out in detail. The formulae by which surface flux values are estimated at each point in model space and time are described, and the assumptions behind their derivation are discussed (as requested by Reviewer 1, Thorndike (1992) is cited at this point, as it contains many of the key assumptions used).

Following the request by Reviewer 2 for a more detailed description of the method application, the calculation of induced surface fluxes is then described in turn for three climate variables (melt onset, downwelling LW and ice thickness), with the aid of a new figure (Figure 5), which shows for each variable the fields of variable bias, surface flux dependence, and induced surface flux bias. The calculation of the ice thickness component is described in particular detail, as it now involves the estimation of thickness biases by category which is nontrivial.

It is noted that in the updated analysis the treatment of the turbulent fluxes is changed; their contribution is no longer neglected. However, they are treated in such a way that their dependence on any climate variable examined cannot be evaluated, save for the ice area.
As a result of this change in emphasis, their omission is no longer justified in this section
(and the corresponding Figure 2 is removed); instead, the implications of their treatment is
discussed in the new Section 6.

**2.5 ISF results**
The presentation of the results of the ISF analysis has been changed in several ways.
Firstly, the ISF biases (with CERES as radiation reference dataset) are presented in the new
Table 1, following the request by Reviewer 1 for more systematic quantification of the
results. The table also shows total ISF bias, total net radiation bias relative to CERES, the
residual between the two, and the CERES-ERAI net radiation bias, in preparation for a
discussion of the relationship between observational uncertainty and ISF uncertainty,
inspired by the request by Reviewer 2 for the ISF residuals to be discussed.

The results are also presented in Figure 6 (equivalent to Figure 6 in the previous version).
This figure is unchanged in essence, but now shows the ensemble mean induced surface
fluxes for the newly-refined Arctic Ocean region (see 2.3 above). Total ISF fluxes are also
shown more clearly, using black bars.

Following the presentation of Table 1 and Figure 6, the significant ISF biases are described
in turn (June surface melt onset, August ice area, early winter ice thickness, winter
downwelling LW). In each case, the bias is now quantified, and an equivalent figure for bias
in sea ice growth and melt is described, following the request by Reviewer 1 to make the
surface flux-sea ice mass balance link more explicit. Internal variability in the ISF biases is
then described, using all 80 ensemble years (Reviewer 2 request). Residuals between total
ISF and total net radiation biases are quantified and compared to observational uncertainty
in net radiation (Reviewer 2 request). The direct effect of observational uncertainty on ISF
biases is described. Potential errors in the ISF biases arising from method assumptions,
discussed in detail in Appendix A, are discussed (Reviewer 1&2 requests).

The discussion of ISF spatial patterns that follows has been left largely unchanged. A new
paragraph has been appended to this, comparing the spatial pattern in total ISF bias to that
in net radiation bias, with the aid of a new figure (Figure 7).

**2.6 Discussion**
Following the suggestion by Reviewer 1, the diagnosis of the thickness-growth and ice
albedo feedbacks from the ISF analysis has been justified in a new paragraph. The resulting
relative contributions (over the course of a year) of the feedbacks and forcings has been
quantified (both in terms of surface flux and sea ice melt/growth bias), thereby justifying a
statement in the abstract that Reviewer 1 had rightly questioned. The word 'forcing' has also
been defined more carefully, as it is different from the most commonly used meaning of this
word in a climate context.

Most of the ensuing discussion is unchanged, except for minor rewordings discussed in
Section 3 below, and a mention of Holland et al (2010), requested by Reviewer 2, at the
point at which it is concluded that the June surface albedo simulation problems are the principal cause of of the low annual mean thickness of HadGEM2-ES. However, two paragraphs have been appended, discussing the implications for the results of firstly the imperfect treatment of turbulent fluxes, and secondly the omission of oceanic heat convergence as an additional potential source of model bias in sea ice growth and melt. In this second paragraph, HadGEM2-ES Arctic Ocean heat convergence is compared to observational estimates, following requests by Reviewers 1&2.

**2.7 Conclusions**

This section has not undergone substantial changes as it is considered that the methodological changes have not resulted in any changes to the conclusions of the study. However, two paragraphs have been reworded for improved clarity.

**2.8 Appendix A (analysis of ISF errors)**

This section is equivalent to Appendix B in the original study, but is substantially larger and is based on several suggestions by Reviewers 1&2 for ways in which errors in the ISF analysis should be quantified. Its conclusions are quoted at the appropriate point in Section 5.

Firstly, the error in quantifying dependence of surface flux on climate variable is discussed. This discussion is motivated by comparison of estimated surface flux fields to actual, as suggested by Reviewer 2. As discussed above, although the new methodology has improved the correspondence some differences remain, which can be categorised into three types. In each case, a likely cause of the difference is stated, and its impact on the ISF biases estimated.

Secondly, the error in characterising induced surface flux bias as a product of variable bias with surface flux dependence is described. In this discussion, higher order derivatives of the surface flux are evaluated. The error inherent in using the model state to evaluate surface flux dependence is calculated using the mixed partial derivative terms (this follows from a point made by Reviewer 2).

**3. List of minor corrections and rewordings**

In the following list, the page and line number references refer to the non-tracked changes version of the manuscript. Each alteration is followed with an indication of whether it was requested by Reviewer 1, 2 or neither (in the last case with additional justification). Alterations which are directly related to the new structure, the new methods, or any of the enhancements listed above, are not listed.

1-16 (abstract) 'Counteracting' removed (R2)

1-22 (Intro) '-' replaced with 'to' (R2)

1-27 Stammerjohn et al cited (R2)

1-28 'whoseloss' replaced with 'the loss of which' (R2)

1-34 'observations' replaced with 'reference datasets' (R2)

2-3 second 'very' removed (considered unnecessary)

2-15 'anomalies' replaced with 'biases' here and subsequently (R2)

4-4 (Model and observations) Reference to errors in atmospheric forcing added (R2)

4-14 'IceSAT' corrected to 'ICESat' and expanded

4-24 Unnecessary reference to ISCCP-D cloud product removed

4-28 'To the authors' knowledge' removed (considered irrelevant)

5-12 (Sea ice and surface radiation evaluation) 'Envisat' corrected to 'the ERS satellite
measurements' (this was previously incorrect)

15-1 (Discussion) Sentences reworded along the lines suggested in the original reviewer
response (R2)

15-20 'in the Arctic' added (for clarification)

17-14 (Conclusion) Paragraph reworded for clarity

17-19 Paragraph reworded for clarity

21-15 Acknowledgement expanded

**Appendix A: Original point-by-point reviewer response**

**A1. Reply to Reviewer Comment 1 (Anonymous)**

We thank the reviewer for taking the time to read our manuscript, and for his/her useful
suggestions for its improvement, which we address inline below. The reviewer's comments
are quoted in italics.

*West et al propose a new analysis framework to understand model biases in Arctic sea ice*
*which they apply to HadGEM2-ES, a model with known biases in sea ice characteristics.*

*The attribution of climate model errors in the sea ice zone is a very important open topic and*
*the paper provides original and likely efficient means to evaluate such errors. The main*
*problem I think is writing, which I found often imprecise, and renders a proper evaluation of*
*the paper difficult.*

*In particular, the methods absolutely require clarification and should use better and simpler*
*terminology. Because I did not fully get the methods, it was thereafter really complicated to*
*follow, in particular the discussion and conclusions.*

*A second requirement to make this paper acceptable is to early on in the result section to explain that the induced surface flux method works - eg. to describe how well the different methods to compute surface flux biases converge. Now this is done here and there, and I have constantly been doubting of the quality of the methods, because of the absence of such evaluation.*

The reviewer is right that the convergence of the different methods, demonstrated in Figure 6, deserves better discussion, and probably quantification, which we propose to carry out by more thorough analysis of the errors of the induced surface flux method in Appendix B, and discussion of these in Section 4, as described below.

Our view is that the spread amongst the different estimates is caused predominantly by observational uncertainty, with errors introduced by the induced surface flux method assumptions relatively small in magnitude by comparison. This is supported by the fact that the difference between the sum of the induced surface flux contributions and each surface flux anomaly is comparable in magnitude to the differences between surface flux anomalies wrt the different datasets (ERAI, ISCCP-FD, CERES).

We think that it would be difficult to show that the induced surface flux (ISF) method works purely by comparison with the direct surface radiation evaluation, because the difference between the different estimates are dominated by the observational uncertainty. We think a better way would be by a more thorough evaluation of the impact of assumptions made by the ISF method in Appendix B. For example:

- Evaluation of errors introduced by the simple model by comparing modelled fields of net radiation to those predicted by the formulae, as suggested by Reviewer 2
- Evaluation of errors caused by ignoring higher-order derivatives, by calculating these terms

The magnitude of these errors would then be compared to the observational uncertainty, as estimated by the difference between the direct radiation evaluations shown in Figure 6.

There is a more fundamental point: the ISF method is not just a way of calculating surface flux anomalies due to a particular process, but also of characterising them, because their definition is to some degree subjective. For example, suppose for the month of May a model shows mean downwelling SW of 300 $Wm^{-2}$, and albedo 0.8, given net SW of 60$Wm^{-2}$, but observational estimates shows mean downwelling SW of 250 $Wm^{-2}$, and albedo of 0.7, giving net SW of 75 $Wm^{-2}$, with a model anomaly of 15 $Wm^{-2}$. Clearly the downwelling SW anomaly induces a positive surface flux anomaly, the albedo anomaly induces a negative one, and the total surface flux anomaly is -15 $Wm^{-2}$. But the exact induced anomalies are subjective. The approach used in the paper is equivalent to multiplying the anomaly in one process by the mean in the other, giving contributions of +12.5 $Wm^{-2}$ and -27.5 $Wm^{-2}$ by the downwelling SW and albedo anomalies respectively.

Hence while it would in theory possible to say whether the *sum* of the ISF contributions was correct (if we knew the exact actual surface flux error), it would not be possible to say whether each individual contribution were correct. The main requirement is that each contribution is physically realistic, and provides useful information.

*A third thing I would have enjoyed to see is a specific discussion of how the ice-albedo and growth-thickness feedbacks can be diagnosed from the method. It is claimed in the abstract that your method can separate these effects, and I am in trouble to see how that statement is presently supported in the text. I can guess feedbacks are acting from Fig. 6, but I think this topic deserves a bit more to support the claim made in the abstract.*

This does require greater justification. The sentence in which the relevant anomalies are identified with the surface albedo feedback and thickness-growth feedback (page 10, line 31) will be expanded accordingly.

The thickness-growth feedback, for example, ostensibly acts by altering the energy balance at the base of the ice; as the ice thickens, the temperature gradient decreases, basal conduction decreases, and the energy balance becomes less strongly negative, so the ice thickens more slowly. But by energy conservation, this process must also have some manifestation in the external fluxes: as the ice is losing energy less quickly, some external energy flux must also have changed. Under the assumptions of the simple model used (similar to Thorndike 1992 as you note below), which ignores sensible heat storage, it is the upwelling LW term that changes: as the ice thickens, its top surface also cools to maintain flux continuity.

Hence any change to the ice energy balance resulting from the ice thickening, and therefore conducting less efficiently, can be diagnosed as the contribution of the ice thickness to the change in upwelling LW radiation.

*I have also not understood why energetic errors of oceanic origin have been ignored from the discussion, especially in the North Atlantic sector of the Arctic - where there is a low bias.*

Energy passing to the ice through the ocean-to-ice heat flux has two main sources: solar input to the ocean, and oceanic heat convergence. The first is implicitly taken account of through the analysis of the effect of ice fraction anomalies on net SW radiation. We make the case that Arctic-wide, the importance of the oceanic heat convergence in driving summer basal ice melt is small by comparison to direct solar input, and this will be more thoroughly justified in the revision, referencing model results by e.g. Steele et al 2010 in addition to the observational references originally included.

However, you are right that the contribution of the oceanic heat convergence should be properly quantified. In the revised version of the paper, we will include estimates of Arctic-wide ocean heat convergence (for model and observations), and set the results shown in Figure 6 in this context.

*Finally, the authors claim in the conclusions that they can "quantify" the origin of errors, but apart from Fig. 6 (which I liked a lot), I did not really see a quantification of the errors. Is that quantification the main point - or is it the consistent comparison of the different sources of error ?*

We do quantify individual contributions to the surface flux biases as shown in Figure 6 for a few illustrative months, in section 4. However, we will examine whether there is scope for a more systematic approach, for example quoting the annual average flux contribution for each state variable in a table.

*Also, it was difficult to ultimately figure out whether biases in external forcings or in the sea ice model are the ultimate cause of the biases. Is your method capable to tell after all ?*

Briefly, the answer is no: the method cannot tell the ultimate cause of the surface flux biases. It is designed to diagnose the proximate cause of the biases.

The induced surface flux (ISF) method, alone, can determine only the first-order cause of the net surface flux bias. The state variables examined (downwelling SW, downwelling LW, melt onset occurrence, ice fraction, ice thickness) affect the surface flux on very short timescales, and are unambiguously properties of the atmosphere (radiation) and sea ice (melt onset, fraction, thickness). Hence the ISF method allows the short-term causes of surface flux bias to be decomposed into those arising from the atmosphere, and from the sea ice.

It's recognised that the causes of biases in the state variables themselves may lie in different systems. For example, ice thickness biases will have some ultimate cause in the atmosphere – indeed, that is one finding of the paper. Conversely, while the case is made in Section 5 that cloud errors are to blame for downwelling LW biases, it is likely that the sea ice simulation will nevertheless have some influence on how this is manifested. However, all these effects act on relatively long timescales, compared to the almost instantaneous timescale on which the state variables affect the net surface flux.

*A last general comment - the logics of the arguments should be better presented.*

We apologise that the presentation of logic is unsatisfactory. This paper has been through several rounds of restructuring as the analysis has developed, and the coherence of argument has probably suffered due to this. We will try to significantly improve this in the revision.

*I am pretty confident that - if these presentation issues are seriously addressed by the team of coauthors, this will make an excellent contribution to their favourite cryospheric journal.*

*A few specific comments.*

*—*

*\* I have tried to understand what the generic approach is.  Here is what I have understood. The present presentation is too lengthy, misses the essential elements and overdiscusses details. A synthetic view is missing. There are three means to evaluate errors in surface energy budget (I have understood two of them)*

*1) The direct computation of surface flux bias, i.e. the difference between simulated and observed surface flux (or one of its components)*

This is correct although we would add the caveat that this is still only an estimate of the actual surface flux bias – the observational uncertainty is very large.

*2) The induced surface flux bias, which is the contribution of bias in a specific variable to surface flux bias, namely calculated as $\Delta Fx = dFx / dx \, \Delta x(mod\text{-}obs)$.*

*To evaluate derivatives, the SEB is simplified using two different approximations during the cold and warm seasons, based on ideas from Thorndike et al 1992.*

*I don't think there is a need to calculate those derivatives in the body of the paper.*

We broadly agree but note the concern by Reviewer 2 that some of the calculations by which the induced surface fluxes are arrived at are incompletely explained. It may be necessary to show at least one derivative, for illustration, with the additional detail that is planned for the revision.

*If the derivatives are well calculated and if the non-linearities are not too important, the sum of $\Delta$ Fx should hopefully approach the surface flux bias.*

*3) The third diagnostic is "the sea ice latent heat flux uptake anomaly implied by the ice volume anomalies relative to PIOMAS".*

*I have tried to figure out what the authors mean, but I did not really managed. The wording is not precise enough for the reader to what is meant by this and what is gained by comparing that to the surface flux biases. I guess "latent heat flux" is confusing in the context of the surface energy budget. But whether that thing is a heat storage anomaly divided by time or something else, I don't know. Maybe an "ice thickness bias converted to Joules" or "an energetic equivalent ice thickness bias" ?*

This is correct. For each month, the modelled field of rate of change of ice thickness is calculated as half the difference between the following and the preceding month. A similar field is calculated for the reference dataset, PIOMAS. The reference field is subtracted from the modelled field to create a model anomaly of ice thickness change. This is then multiplied by ice density, specific latent heat of fusion, reversed in sign, and divided by the number of seconds in a month, to create an equivalent sea ice latent heat storage anomaly in Wm-2.

For the reasons discussed in Section 2.3, the surface heat flux is viewed as the main source of this latent heat storage. It's noted however that it would be useful to provide an estimate of modelled and observed oceanic heat convergence which provides an additional input to the latent heat storage.

We will clarify this point in the revised manuscript.

*Besides an explanation of what it means, we would need an explanation of what should be taken from that diagnostic.*

*It is important to clarify this point because a lot of the argumentation was based on that.*

*\* The two methods to compute the surface flux derivatives is called "a model". I think it is a*
*"computation method". It is actually inspired from Thorndike et al (1992) – which should be*
*acknowledged - and maybe from earlier works in EBMs. What you are doing is to derive the*
*surface energy budget wrt anything.*

The 'model' versus 'computation method' is an interesting distinction – our interpretation is
that it hinges on wither the formulae are viewed as a way of calculating induced surface
fluxes (model) or characterising them (computation method), as discussed above. As in our
view both are valid interpretations, either phrase might be more appropriate depending on
the circumstances.

Thorndike et al will be cited.

**A2. Reply to reviewer comment 2 (Francois Massonnet)**

We thank Francois Massonnet for his helpful and thorough review of our manuscript. Below,
we address in turn his major and minor comments inline, which are quoted in italics.

**Major comments**

1) *It is not always easy to follow the authors' methodology. I see the general idea behind*
*the approach: expressing the net flux to the ice as a function of state variables, then*
*linearizing around a reference state to obtain the flux bias resulting from the bias in*
*one of the model components. However, I could surely not reproduce the results*
*myself, just based on the text. I appreciate the efforts to publish the code in*
*Supplementary Material, but the text itself should have all elements. For example, I do*
*not understand how the bias in F_sfc attributed to error in melt onset is derived (i.e.,*
*from eq. A6 to 4). Furthermore, the attribution of flux error to melt onset error*
*seems to not be a function of the melt onset itself, but rather a function of*
*concentration difference. This is confusing: melt onset is defined by the time of the*
*day where surface melting commences, and the right hand side of Eq. 4 does not*
*display surface melting terms. At some point it came to my mind that the authors were*
*perhaps using "melt onset" for "ice retreat", but I'm not sure. In all cases, this is*
*confusing.*
The definition of the state variable 'melt onset occurrence', and its relation to the net surface
flux, is not very clearly explained in the paper, and certainly requires expansion. This will be
altered in the revised version of the paper, and we will ensure more generally the replicability
of the calculations of the induced surface fluxes (for example, noting the use of local ice and
snow thicknesses as you suggest below). However, a brief explanation of this particular
component, melt onset occurrence, is also provided here. Very simply, its purpose is to
capture the effect of meltpond formation on the surface flux.

HadGEM2-ES parameterises the effect of meltponds by reducing surface albedo linearly from 0.8 to 0.65 as the surface temperature goes from -1°C to 0°C, after Curry et al (2001). Because we have daily surface temperature fields, we can judge for each modelled year which day 'melt onset' – defined as the day surface temperature first goes above -0.5°C – occurs. Comparison of these dates to the observational SSMI estimates referenced in the paper shows that modelled melt onset occurs, on average, 20-25 days earlier across most of the Arctic Ocean in the model than in observations.

In the induced surface flux analysis, we examine the effect on modelled surface flux of the melt onset process occurring at the wrong time of year. The relevant state variable here is 'melt onset occurrence', which takes the value 0 or 1 depending on whether a grid cell on a particular day has yet exceeded -0.5°C (model definition) or whether a liquid water microwave signature has been detected (observational definition). In a similar way to the other state variables used in the paper, the observations are averaged over the period 1980-1999 to obtain a daily climatology of melt onset occurrence. For each modelled year, this climatology is subtracted from the modelled melt onset occurrence fields to obtain a modelled melt onset anomaly. This anomaly is then multiplied by the relevant partial derivative – in this case, (downwelling SW) * (cold snow albedo – melting snow albedo) to produce the induced anomaly in net SW.

> 2) *Besides the need for clarity in the methodology, a key question is to what extent the assumption of linearity holds, in particular for what bias range Eqs 3 and 4 would be valid. Will the methods work for models with very large biases? Another point is that this linearization involves the use of Eqs 1 and 2, that are themselves derived using linearity assumptions. I trust that the approach is valid, because the sum of individual contributions (Fig. 6) seems to match the flux errors from datasets and from volume estimates, but a quantification of this match should be done (perhaps by calculating residuals). Overall I find that the authors have not discussed the validity of this assumption, and this is critical given how non linearly the ice behaves.*

You are right that this needs to be quantified. For some of the state variables the dependence of surface flux is linear, and the second partial derivatives go to zero (e.g. all state variables in the melting season, and downwelling LW in the freezing season). However the dependence of surface flux on ice and snow thickness is nonlinear and it would be useful to examine the circumstances in which higher derivatives are important.

As you mention, additional assumptions are made in deriving equation (1): linearity of upwelling longwave dependence on surface temperature, and uniform conduction of heat within the ice. We will try to quantify the impact of these also by comparing actual net surface flux fields to predicted fields, as you suggest below in point 5).

> 3) *I would also like to see if the method is robust to internal variability. Could the authors take one or several of the four other members of the HadGEM2-ES model and run the same analysis? In other words, is the 1980-1999 period long enough to identify and attribute the biases?*

Analysis of other ensemble members would be a valuable enhancement of the study. For the revised version of the paper, we plan to carry out the same analysis on the other three ensemble members, and to quantify the consistency with the results from the first member.

4) *The authors have not cited an important study: Holland et al., 2008 (doi:10.1007/s00382-008-0493-4). In that study, the inter-model scatter in the sea ice mass budget (present-day conditions) is shown to be explained by the way models absorb shortwave radiation. This is directly relevant to the study here, and I think the authors should go through the Holland et al. study to position their results with respect to theirs. In particular the claim that turbulent fluxes are of relatively minor importance relative to radiative fluxes in setting the surface energy balance (Fig. 2, and p. 5 line 11-16) should be put in perspective with that study. As the Arctic sea ice mean state changes, turbulent fluxes appear to be of increasing importance.*

Holland et al (2008) show annual sea ice melt rates to be strongly correlated with summer net SW across the CMIP3 ensemble. The causality here could go in either, or most likely both, directions. This appears to be consistent with the finding in the current study that the excessive sea ice melt in HadGEM2-ES is driven by surface albedo and net SW issues. This will be referenced.

The neglecting of turbulent fluxes is a shortcoming of our study. As you point out, while they are comparatively small in an absolute sense, they may nevertheless be important in driving future changes. In a similar way, model anomalies in turbulent fluxes may be of comparable size to those in radiative fluxes even if the model absolute values are much smaller. We will expand on this point in the Discussion.

5) *The authors should prove, with a figure, that the model developed in the appendix is good enough to do the investigations. Could they plot, for one or several grid cells and one or several freezing seasons, the reconstructed flux F_sfc (Eq. A4) and the actual flux from the model? A quantification of the correspondence would be a plus.*

This would also be a valuable exercise. Actual and calculated modelled fields will be compared for a few sample years, and the correspondence described, quantified and illustrated. The most appropriate place for this would probably be the discussion of errors in Appendix B.

6) *Nothing is said about the treatment of snow in the HadGEM2-ES model. How many snow layers are there, what is snow conductivity, etc.?*

There is only one snow layer, and the conductivity is 0.33 Wm-1K-1. Like the ice, the snow has no heat capacity. It should be made clear, however (here and in the revised version of the paper) that sensible heat storage is parameterised in the top 10cm of the snow-ice column during surface exchange calculations, to aid stability.

7) *The authors repeatedly use the word "anomaly" to describe the difference between modeled and reference quantities, but I would avoid this word and use "bias" or "error" instead. To me, an "anomaly" is used to described the deviation of a signal with respect to its own mean*

We have some concern is that use of the word 'bias' might suggest that HadGEM2-ES is being evaluated with respect to the 'truth', but most datasets used are only very rough approximations to this. We will change 'anomaly' to 'bias', but clearly define at the outset the meaning of the word 'bias' for the purposes of the paper – the difference of the model relative to a particular observational estimate.

8) *I'm unclear about whether ocean surface temperature biases are accounted for in the analysis. From Fig. 6, it looks like they are not. On the other hand, p. 5 lines 6-10 seem to suggest that the Arctic Ocean is critical in setting the ice energetic balance (and this is also seen in Keen et al (https://link.springer.com/article/10.1007/s00382-013-1679-y, their Fig. 4). So, I'm puzzled: is the contribution of oceanic surface temperature bias taken into account or not in the analysis?*

It is true that the ocean contributes a significant amount of heat to the ice in the summer. However, we make the case in our study that the major part of this heat comes from direct solar heating of the ocean, an effect which is taken account of through analysis of the effect of ice fraction on the net SW bias. This case will be strengthened in the revision by citing evidence from models (e.g. Steele et al, 2010) in addition to the evidence from observations already referenced.

This is also relevant to the minor comment at 5-6/10 below.

**Minor comments**

*1-18 - "countered by a counteracting" is a bit odd.*

'Counteracting' is superfluous and will be removed.

*1-24 - "from 1986-2015" –> from 1986 to 2015*

Change will be made as suggested.

*1-28 - Along with an earlier melt onset date, you can mention that freeze up has been delayed: Stammerjohn et al., 2012, their Fig.2 (doi:10.1029/2012GL050874)*

This will be done.

*1-29 - "whoseloss" –> "the loss of which"*

Change will be made as suggested.

*2-1 - Evaluation against observations of volume is quite impossible (even extent*
*observations are not direct observations),  so I would use "observational or reanalysis*
*reference datasets"*

Change will be made as suggested.

*2-18 Instead of "anomalies" I would use "biases"*

See response to point 7) above.

*3-24 The period 1980-1999 is used for evaluation, because it "predates the rapid sea ice*
*loss".  Why is it a problem to have a period with strong trend in the analysis? Is it expecting*
*that the SEB would change too rapidly during a period with strong trends? Would the*
*analysis be robust if the model output was evaluated on a distinct and later period (2000-*
*2015 for instance, using historical + RCP8.5 runs). Please elaborate.*

The trend itself is not problematic. Our motivation for using this period was to evaluate the
model on a 'reference' time period at least partially independent of the time period that is
usually used to evaluate sea ice trends. We will make this clear in the revised version.

*3-35 Reanalysis data also suffer from biases because of errors in atmospheric forcing, this*
*could be stated as well.*

This will be stated.

*5-11/16 Can the authors explain exactly what they mean by "Heat flux due to snowfall". The*
*presence of snow affects heat conduction fluxes and acts to reduce bottom growth, is that*
*what the authors are talking about?*

The presence of snow matters only because it takes energy to melt it; snow falling on ice
changes the enthalpy of the snow-ice system. A 3m column of bare fresh ice at 0°C, for
example, will take ~9.2 x $10^8$ Jm$^{-2}$ to melt it. If 50cm fresh snow falls on the ice, the
combined snow-ice column will take ~9.8 x $10^8$ Jm$^{-2}$ to melt it. Hence the falling of snow on
ice represents a transfer of negative latent heat from the atmosphere system to the snow-ice
system, which must be taken account of in calculating the total surface flux. We will try to
explain this more fully in the revised version of the paper.

*5-26 I assume h_I and h_s refer to in-situ / actual thickness (this is the one that matters for vertical thermodynamics). It would be good to mention that here, as there is usually a lot of confusion between that quantity and the grid cell average thickness.*

Yes, this is the case and that will be clarified.

*5-24 Eq. 1: Maybe I missed it, but what is the value for ice albedo? Does albedo depend on the ice state?*

The ice albedo used is 0.61. This does not depend on ice thickness, but falls linearly to 0.535 as ice surface temperature rises from -1°C to 0°C. This information will be added to the text.

*5-27 The subscript for snow thickness is "S" here while it is "s" elsewhere*

Change will be made as suggested.

*5-28 The symbol for albedo is \alpha_I while in the equation it's \alpha_ice*

Change will be made as suggested.

*5-30 Eq. 2: the big "dot" is a bit disturbing, it makes me think at a scalar product. I would use a simple dot or no dot at all.*

Change will be made as suggested.

*5-6/10 The sentence "Because of this, although advection-derived ocean heating..." is unclear to me. First, can you demonstrate the oceanic heat convergence (that is not accounted for in your framework) is a small contributer to volume changes? Second, I do not follow the logical articulation with the next sencen "Hence the surface energy...". Please clarify. In the same line, reading the recent paper by Lei et al. (2018, doi:10.1002/2017JC013548) could be useful to add up to the discussion.*

Regarding your first question, Reviewer 1 also suggested that it would be sensible for the oceanic heat convergence to be properly quantified, and this will be done in the revised version.

Regarding your second question, see first our response to your major comment 8) above. We agree our wording here is confusing. The case we are making is probably best illustrated by this schematic:

[Figure]

The point made is that the source energy for the ocean-to-ice heat flux derives, in the main,
from the surface heat flux (specifically, solar heating in summer), and not from oceanic heat
convergence, over most parts of the Arctic Ocean (clearly there are regions where this is not
true, e.g. near the ice edge in the Atlantic sector). Therefore the surface flux analysis
(specifically, the effect of ice fraction anomalies on net SW) implicitly accounts for a large
part of the ocean-to-ice heat flux.

As we mention above, additional evidence will be cited for this in the revision, as well as
rewording this sentence.

Thank you for drawing our attention to Lei et al (2018), which draws together a very wide
range of observational data to investigate mechanisms of sea ice growth and melt. If we
have understood this study correctly, it deduces a strong role for direct solar heating in
driving summer sea ice basal melting by noting an association between areas of low
summer sea ice concentration and high early autumn oceanic heat fluxes (as measured by
ice mass balance buoys). Hence this would also be a valuable study to quote in this context.

*6-1 Please give the albedo values used.*

These will be provided in the revised version: 0.535 for melting ice, 0.61 for cold ice, 0.65 for
melting snow, 0.8 for cold snow.

*6-3 "summarises" –> "summarise"*

Change will be made as suggested.

*6-10 Eq.3: please describe the meaning of the terms of the equation.  In particular,  what is*
*h_l_eff?  It  is  necessary  to  have  this  information  in  the  text  somewhere.*

These will be described fully.

*6-23 The partial derivatives are to be evaluated at a reference state, and I understand here that a mid-point between observations and model is taken ("Where observational datasets were available, the reference quantities in the partial derivative fields were calculated as model-observation means"). The authors should explain why it was done this way. I assume that the reconstructed flux error would be mathematically closer to the actual error than if the reference was taken as either the model or the osbserved value.*

I don't think that's the case. For any function $f$, evaluated at two values $x_1$ and $x_2$, if we try to approximate $f(x_2)-f(x_1)$ using the first term of the Taylor series evaluated at some point $\lambda x_1 + (1-\lambda)x_2$, where $0 \leq \lambda \leq 1$, the coefficient of the second Taylor series term is minimised when $\lambda = 1/2$, i.e. at the midpoint of $x_1$ and $x_2$. Hence evaluating the partial derivatives at the model-observation midpoint provides our best guess. We will briefly note our reasoning for using this in the revision.

*6-30 The paragraph starts by saying that 4 ensemble members were run, but Fig. 3 only shows one. Can you clarify?*

Only one ensemble member is used for the analysis – this will be clarified, although as indicated above reference will be made in the revised version to results for the other three members.

*8-28 The word "save" should be removed, I think*

It would probably be best replaced by 'except for'.

*10-25/28 Can you go a bit more quantitative here? From Fig. 6, the residual of the analysis can be calculated as the sum of individual contributions (the stars) and the actual flux error.*

But what is the 'actual flux error'? The surface flux anomalies wrt ERAI, ISCCP-FD, CERES cannot be regarded as such because of the observational uncertainties – this is clear, because the difference between the sum of the contributions and each surface flux anomaly is comparable in magnitude to the differences between surface flux anomalies wrt the different datasets. Each is only an estimate of the actual flux error, which cannot be exactly known – it is equally possible that the sum of the contributions is a more accurate estimate than any.

We can provide the residual of the analysis for completion with respect to ERAI, ISCCP-FD or CERES – but the actual numbers will differ greatly depending on which dataset is used.

*10-31/32 The surface albedo feedback is not just a sea ice concentration thing. The melting of snow, the thinning of the ice are also key players in the surface albedo reduction, even*

*though ice concentration remains unchanged. Wouldn't it make more sense to include these factors as well in the definition of surface albedo feedback?*

It is true that the surface albedo feedback also includes these effects. However, it would not be possible to include the effect of snow melting because of the lack of reference dataset. The effect of ice thickness on albedo is not actually modelled by HadGEM2-ES – the albedo switches abruptly to the open ocean value when the sea ice thickness falls to zero. Hence there would be two separate effects to estimate here: the direct effect of ice thickness anomalies on albedo, and the effect of HadGEM2-ES not modelling this. Given this, as well as large uncertainties in observations of the link between albedo and ice thickness for thin ice, we think this effect is outside the scope of the study. However, it will be mentioned as a possible additional contributing factor to the summer net SW bias.

*11-15 The sentence "Hence the melt onset anomaly, acting alone, would induce a seasonal cycle of sea ice thickness both lower, and more amplified, than that observed..." is unclear, especially regarding the "lower" part. Can you please rephrase?*

How would the following be:

'Hence the melt onset anomaly, acting alone, would induce a seasonal cycle of sea ice thickness lower in the annual mean, but also more amplified, than that observed, because the surface albedo and thickness-growth feedbacks act to translate lower ice thicknesses into faster melt and growth. For similar reasons, the downwelling LW anomaly, acting alone, would induce a seasonal cycle of sea ice thickness higher in the annual mean, and also less amplified, than that observed.'

*12-1 "concludethat" –> conclude that*

Change will be made as suggested.

*Fig. 3. I'm puzzled by panel (a). Sea ice extent seems small. Is that because the domain "Arctic Ocean" is restricted to the seas of Fig. 1? In other observational records, like NSIDC, winter sea ice extent is more in the 14-16 million km2 range.*

Yes, the extent is calculated over only the Arctic Ocean domain, like other variables in this paper, and so the winter extent appears much lower than in the well-known NSIDC figures. We think that this is appropriate, because much of the winter variability in whole-Arctic sea ice extent is due to processes in the subpolar seas, which are not relevant to the Arctic Ocean process analysis in this study.

**References**

[revised manuscript text omitted]

Model anomaly in October-April thickness change (m)

Model anomaly in October-April thickness change (m)

**Figure 34. HadGEM2-ES 1980-1999 model bias  in ice thickness change from October-April compared to (a) PIOMAS 1980-1999; (b) Envisat 1993-2000; (c) submarine regression analysis 1980-1999. Differences are taken as model-observation so that areas of green (purple) correspond to areas where the HadGEM2-ES model simulates too much (not enough) sea ice growth through the winter**

[Figure]

[Figure]

Figure 45. (a) Downwelling SW, (b) upwelling SW, (c) net down SW, (d) downwelling LW, (e) upwelling LW, (f) net down LW, for HadGEM2-ES 1980-1999 over the Arctic Ocean region, compared to CERES 2000-2013, ISCCP-D 1983-1999 and ERAI 1980-1999. For all fluxes, a positive number denotes a downward flux and vice versa. Maps of flux bias anomaly relative to CERES are shown for downwelling SW in May, upwelling and net down SW in June, and downwelling and net down LW in February.

[Figure]

**Figure 5. Demonstrating the calculation of fields of surface flux bias due to model bias in melting surface fraction (a-c), downwelling LW (d-f), category 1 ice thickness (g-i) and category 5 ice thickness (j-l). The left-hand column shows model bias in each variable; the middle column the local rate of dependence of surface flux on each variable as calculated above; the right column the induced surface flux bias, calculated as the product of these two fields.**

[Figure]

~~**Figure 6. Surface flux anomaly induced by model anomalies in ice fraction, melt onset occurrence, ice thickness, downwelling SW and downwelling LW respectively, for the Arctic Ocean region in HadGEM2-ES, 1980-1999, as estimated by the simple models described in Section 2.3. For each month, induced anomalies are estimated using in turn CERES, ISCCP-FD and ERAI as radiation reference datasets, from left-right. Sea ice latent heat flux uptake anomaly relative to PIOMAS is indicated in black. Net radiative flux anomalies relative to CERES, ISCCP-FD and ERAI are indicated in brown. Spatial patterns of induced surface flux anomaly for four processes in key months, with CERES as reference dataset, are displayed beneath.**~~

[Figure]

**Figure 6. Surface flux bias induced by model biases in ice fraction, melt onset occurrence, ice thickness, downwelling SW and downwelling LW respectively, for the Arctic Ocean region in HadGEM2-ES, 1980-1999, as estimated by the simple models described in Section 2.3. For each month, induced surface flux biases are estimated using in turn CERES, ISCCP-FD and ERAI as radiation reference datasets, from left-right. Sea ice latent heat flux uptake bias relative to PIOMAS is indicated in black. Net radiative flux biases relative to CERES, ISCCP-FD and ERAI are indicated in brown. Spatial patterns of induced surface flux bias for four processes in key months, with CERES as reference dataset, are displayed beneath.**

[Figure]

[Figure]

Figure 7. Surface flux anomalies caused by anomalies in external forcings, and to feedbacks due to anomalies in the sea ice state, represented as stacked filled regions. All values shown are means across radiation datasets shown in Figure 4; summer radiative flux anomalies are not plotted due to very large spread among datasets.

[Figure]

**Figure 7. Comparing fields of total ISF bias to net radiation bias relative to CERES for each month of the year, for the four historical members of HadGEM2-ES, 1980-1999.**

[Figure]

**Figure 8. Frequency distributions of (a) October-April cloud liquid water percentage in HadGEM2-ES compared to MODIS observations, for the Arctic Ocean region; (b) December-February surface net downwelling LW in HadGEM2-ES in the SHEBA region, compared to the values observed at SHEBA.**

[Figure]

**Figure A1. Illustrating approximated (left) and actual (centre) model net surface flux, as well as the approximation error (right), in (a-c) February; (d-f) May; (g-i) July, for the period 1980-1999 in the first historical run of HadGEM2-ES.**

---

## Author Response (AR3)

Authors' response to third set of reviews of 'Induced surface fluxes: a new
framework for attributing Arctic sea ice volume balance biases to specific
model errors'

**4 Inline response to reviews**

Reviews are in blue; our responses are in black.

--Reviewer 1---

Dear authors,

I appreciate the efforts you made to address my comments.

The reasoning is much clearer. I still found parts of the paper a bit lenghty, but overall
understandable.

The paper is very original and useful, you have a nice, well illustrated story, and therefore, I
recommend publication.

I'd recommend to make sure figures are uploaded with correct resolution. I doubt of the use of
Figure A1 in the paper. I would put it in supplementary material.

We thank the reviewer for their kind comments, and for their patience during the lengthy review
process. For the final version of the paper, the figures will be uploaded in EPS format, and will be of
much higher quality. We are happy to put Figure A1 in supplementary material, as long as Reviewer
2, who requested this figure, is also happy with this.

Congratulations!

--Reviewer 2—

Once again, I want to underline the great effort made by the authors to take my comments into
account. As far as my comments are concerned, the paper is good to go for publication. I have noted
the re-writing of Section 4, with a step-by-step illustration of the method with two processes. This
will be of much help for those interested to apply the framework in the future, for the authors' own
benefit.

I have made a couple of minor comments while reading the new version of the manuscript. Note
that the page/line numbers below refer to the track-change version of the manuscript that was
appended after the response to reviews.

Title: Now the focus seems to be on sea ice mass balance biases, not on volume biases anymore. I
understand the initial motivation behind this move: it is mass and not volume that is proportional to
the heat required to melt the ice out. I'm a bit skeptical about this choice though, because there are
no proper observational estimates of sea ice mass (that I know), so that it is implicitly assumed that
modeled biases in sea ice density (or parameter values used) are not introducing further uncertainty
in the estimated mass bias. In addition, throughout the text and figures, biases are still expressed in terms of thickness not mass. I do not understand why the focus has been changed and would stick to volume biases. Perhaps I'm missing something but then would appreciate a better justification. I understand that there is a big momentum with the sea ice mass budget intercomparisons within SIMIP that could justify this change, but it's not a good scientific reason I think.

We think this point is a very valid one. Volume balance is a much more accurate description of the focus of the study than mass balance, and we have changed all instances of 'mass balance' accordingly. We have added a note to the effect that an ice density of 917 kgm$^{-3}$ (the value used in HadGEM2-ES) is assumed.

1/17 "... along with the roles of model bias in variables external to the sea ice state...": Do you refer to atmospheric and ocean biases? If so, state it explicitly.

The variables so described include the atmospheric and ocean biases, but also the bias in surface melt onset. We agree the wording here is confusing as this last variable is effectively part of the sea ice state. We have changed this to 'variables not directly related to sea ice volume'.

1/19 "There is mention of sea ice volume biases here in the abstract, but the title now states mass balance biases". It should be made clearer biases in which state variable are investigated (as written above I would suggest to stick to volume).

Changed to volume balance as suggested.

1/23 The sentence on observational uncertainty is much welcome and will be appealing to many readers I think.

3/1 Instead of "observed or reference" I would write: "Hence, using reference datasets (observational estimates or reanalyses)". A "reference" is anything that one can compare the model results to, regardless of its origin.

Change made as suggested.

6/38 A "thin bias... in thickness" sounds a bit strange. How about "low bias in thickness"?

Change made as suggested.

7/8 "a mass balance bias of 38 cm" is typically something that will confuse many readers, even though a few modellers might understand what is meant.

This sentence has been reordered, so that this phrase is explained after being introduced.

7/31 Units Wm-2: "-2" should come as an exponent

This has been corrected.

8/18 "we briefly digress" and the whole paragraph: this really sounds like a paragraph added because a reviewer wanted further information and you did not know where to add this information. Try being a bit less obvious :-)

We have removed this qualification.

- Regarding PIOMAS assimilation, the authors are right that no SSS data is assimilated.

Fig. 6 still has "anomaly" near the colorbars

We have changed this to 'bias', and similar occurrences in Figures 2 and 3.

**List of changes to the document**

All page and line numbers refer to the tracked changes version of the document, appended below.

Page 1, line 2: 'mass balance' changed to 'volume balance'

(Subsequent instances of this change are not listed)

Page 1, line 15: 'external to the sea ice state' changed to 'not related to the sea ice volume'

Page 2, line 9: Associated with the change in focus to volume balance, a qualification related to the sea ice density is added.

Page 2, line 24: 'Observed or reference datasets' changed to 'reference datasets (observational estimates or reanalyses)

Page 6, line 13: 'thin model bias' changed to 'low model bias'

Page 6, line 17: sentence re-ordered to clarify the meaning of a volume balance bias

Page 7, line 1: Assumptions regarding ice density are described.

Page 7, line 29: 'briefly digress from the radiation evaluation, to' removed

Page 33, figure 2: instances of 'anomaly' changed to 'bias'

Page 35, figure 3: instances of 'anomaly' changed to 'bias'

Page 38, figure 6: instances of 'anomaly' changed to 'bias'

Page 43, figure A1: Figure removed and placed instead in supplementary material.

[revised manuscript text omitted]

---

## Author Response (AR4)

**1** Final authors' response**

- 2 The authors thank the reviewers, and the topical editor, for their hard work in examining, and
- 3 suggesting significant improvements to, this study. As the topical editor requested the consideration of
- 4 additional changes prior to submitting the final version of the study, the final version is significantly
- 5 different in wording in many places to the previous version. A tracked changes version is appended
- 6 below this document, and the changes are briefly summarised here.
- 7 Firstly, the use of 'volume balance' rather than 'mass balance' is explained (Page 2, line 6).
- 8 Secondly, the reviewers and editor are now thanked in the Acknowledgements (Page 26, line 18). We
  9 apologise for this careless omission.
- 10 Finally, following a suggestion by the editor, a member of the author team reviewed the paper again,
- 11 to identify areas where clarity could be further improved. This has resulted in a large number of
- 12 changes, particularly in Sections 1 and 3. In most cases, these involve altering the order in which
- 13 information is presented for example, starting paragraphs with the key point they are making and
- do not affect the scientific content of the document. However, in some cases, information is actually
- removed, having been judged unnecessary, specifically: justification for period of analysis in section 2
- 16 (page 4, line 18); constraint of submarine observations of ice thickness to be seasonally symmetric in
- 17 section 3 (page 6, line 25); description of downwelling SW biases in section 3 (page 7, line 33).
- 18 A tracked changes version of the document follows below.

**Induced surface fluxes: A new framework for attributing Arctic sea ice volume balance biases to specific model errors**

3 Alex West1, Mat Collins2, Ed Blockley1, Jeff Ridley1, Alejandro Bodas-Salcedo1

- 1Met Office Hadley Centre, FitzRoy Road, Exeter, EX1 3PB
- 5 2Centre for Engineering, Mathematics and Physical Sciences, University of Exeter, EX4 4SB, UK

6 *Correspondence to*: Alex E. West (alex.west@metoffice.gov.uk)

[revised manuscript text omitted]

- 1 downwelling SW during the summer, (Figure 4a), suggesting that the a negative model surface albedo is biased
- 2 lowbias. The effect is that modelled netNet downward SW flux is too large with respectshows a high model bias
- 3 relative to all observational datasets in May and June, and with respectrelative to some in July and August.
- 4 Relative to CERES, the May downwelling SW bias displays no clear spatial pattern over the Arctic Ocean
- 5 (Figure 4a), but the June upwelling SW bias, and hence the net SW bias, tend to be somewhat higher in
- 6 magnitude towards the central Arctic (Figure 4b-c).
- 7 As the The June net SW bias is likely to result from a surface albedo bias we discuss the parameters affecting is
- 8 likely to be associated with a bias in surface albedo over sea icemelt onset in HadGEM2-ES: ice fraction, snow
- 9 thickness and surface melt onset. Ice fraction has already been evaluated and for snow thickness no reference
- 10 dataset is available; however, surface (Figure 5). Surface melt onset-can be evaluated using satellite observations
- 11 (Figure 6). We define the date, associated with the formation of melt onset for any grid cell as the first day on
- 12 which the surface temperature exceeds 1°C (varying this threshold by 0.5°C in either direction changes the date
- 13 in only a small minority of grid cells). The average date of melt onset as estimated by this method (Figure 5a) is
- 14 then compared to that measured by the satellite derived dataset described in Section 3 (Figure 5b), with model
- 15 bias shown in Figure 5c. Large spatial variability is evident in the observations. Melt onsetlow-albedo melt-
- 16 ponds, occurs in early to mid Mayaround mid-June in the Ccentral Arctic (although earlier around the Arctic
- 17 Ocean coasts, but much later in the Central Arctic, around mid June.), as measured by satellite observations. In
- 18 contrast, the HadGEM2-ES simulates surface melt onset date is in mid- to late May across the Arctic Ocean,
- 19 without the strong gradients seen in the observations. This would cause a bias in surface albedo, and hence net
- 20 SW, bias with a strong maximum in the cCentral Arctic, similar to that consistent with the spatial pattern of the
- 21 net SW bias discussed above. HadGEM2-ES parameterises the effect of melt-ponds by lowering sea ice albedo
- 22 as surface temperature reaches the melting point; in the comparison above, melt onset date in the model is
- 23 defined as the first day on which the surface temperature exceeds -1°C. Varying this threshold by 0.5°C in either
- 24 direction changes the date in only a small minority of grid cells.
- 25 We now return to the LW radiation evaluation to evaluate LW-fluxes. Fluxes of longwave (LW) radiation (both
- 26 downwelling and upwelling) are lower in magnitude in HadGEM2-ES throughout the winter than in all
- 27 observational datasets (Figure 5d-f). For downwelling LW, the mean model biases from December-April are -
- 28 16, -22 and -40 Wm-2 for ERAI, CERES and ISCCP-FD respectively; for there is no indication from in situ
- 29 studies as to where the true model bias may lie, as in situ studies of downwelling LW measurement have shown
- 30 underestimation in-by CERES and overestimation by ERAI and ISCCP-FD. For upwelling LW, the biases are
- 31 11, 16 and 18 Wm-2 for CERES, ERAI and ISCCP respectively. Because the downwelling LW biases vary more
- 32 than the upwelling LW biases, there There is uncertainty in inferring a model bias in net downwelling LW;
- 33 ISCCP suggests, with negative biases suggested by the satellite datasets but a large model bias of  $22 \text{ Wm}^2$ ,
- 34 CERES a smallerneutral bias of 11 Wm2, whileby ERAI suggests a. The bias of only 1 Wm2. 
[revised manuscript text omitted]
 | Downwelling | Ice       | Ice  | Melt onset | Total     | Radiative | ISF      | CERES-ERAI    |
|-------|-------------|-------------|-----------|------|------------|-----------|-----------|----------|---------------|
|       | SW          | LW          | thickness | area | occurrence | induced   | flux bias | residual | net radiation |
|       |             |             |           |      |            | surface   |           |          | difference    |
|       |             |             |           |      |            | flux bias |           |          |               |
| Jan   | 0.0         | -6.5        | -4.3      | 2.7  | 0.0        | -8.1      | -1.6      | 3.3      | -12.2         |
| Feb   | 0.0         | -4.8        | -2.8      | 2.3  | 0.0        | -5.3      | -10.4     | 4.6      | -12.5         |
| Mar   | 0.1         | -3.8        | -2.0      | 2.0  | 0.0        | -3.7      | -10.5     | 5.9      | -12.0         |
| Apr   | 0.4         | -4.4        | -1.4      | 1.4  | 0.2        | -4.2      | -12.2     | 7.6      | -9.7          |
| May   | 2.1         | -4.8        | -0.6      | 0.0  | 0.1        | -3.2      | -3.7      | 0.5      | -3.5          |
| Jun   | -8.3        | 7.4         | 0.0       | 1.7  | 11.4       | 12.2      | 27.8      | -15.4    | -4.2          |
| Jul   | -13.6       | 8.0         | 0.0       | 3.7  | 3.3        | 1.4       | 5.1       | -3.9     | -16.9         |
| Aug   | 0.5         | -3.3        | -0.1      | 9.6  | 1.9        | 8.5       | 8.6       | -0.0     | -12.9         |
| Sep   | 3.3         | -7.1        | -0.7      | -0.7 | 0.0        | -5.2      | -0.2      | -4.8     | -2.4          |
| Oct   | 0.9         | -5.7        | -4.2      | -1.8 | 0.0        | -10.8     | -14.4     | 3.4      | -4.6          |
| Nov   | 0.0         | -5.4        | -8.3      | -0.3 | 0.0        | -14.0     | -21.6     | 8.1      | -11.7         |
| Dec   | 0.0         | -6.4        | -6.4      | 2.4  | 0.0        | -10.4     | -14.9     | 4.1      | -12.2         |

2 Table 1. Surface flux biases induced by model bias in 5 different variables in HadGEM2-ES (Wm-2), with

3 CERES used as reference dataset for the radiative components. Total ISF bias and total net radiative flux

4 bias relative to CERES are shown for comparison, as well as their residual; the difference between net

5 radiative flux bias as evaluated by CERES and ERAI is also shown. A positive number denotes a

6 downwards flux, and vice versa.

2 Figure 1. The Arctic Ocean region used in the analysis, defined as the area enclosed by the Fram Strait, Bering Strait

Figure 1. The Arctic Ocean region used in the aand the northern boundary of the Barents Sea.